# The nucleosomal acidic patch relieves auto-inhibition by the ISWI remodeler SNF2h

Nathan Gamarra[1,2], Stephanie L Johnson[1], Michael J Trnka[3], Alma L Burlingame[3], Geeta J Narlikar[1]*

[1]Department of Biochemistry and Biophysics, University of California, San Francisco, San Francisco, United States; [2]Tetrad Graduate Program, University of California, San Francisco, San Francisco, United States; [3]Department of Pharmaceutical Chemistry, University of California, San Francisco, San Francisco, United States

**Abstract** ISWI family chromatin remodeling motors use sophisticated autoinhibition mechanisms to control nucleosome sliding. Yet how the different autoinhibitory domains are regulated is not well understood. Here we show that an acidic patch formed by histones H2A and H2B of the nucleosome relieves the autoinhibition imposed by the AutoN and the NegC regions of the human ISWI remodeler SNF2h. Further, by single molecule FRET we show that the acidic patch helps control the distance travelled per translocation event. We propose a model in which the acidic patch activates SNF2h by providing a landing pad for the NegC and AutoN auto-inhibitory domains. Interestingly, the INO80 complex is also strongly dependent on the acidic patch for nucleosome sliding, indicating that this substrate feature can regulate remodeling enzymes with substantially different mechanisms. We therefore hypothesize that regulating access to the acidic patch of the nucleosome plays a key role in coordinating the activities of different remodelers in the cell.

DOI: https://doi.org/10.7554/eLife.35322.001

*For correspondence: Geeta.Narlikar@ucsf.edu

## Introduction

Eukaryotic genomes are packaged into chromatin, enabling large amounts of DNA to fit into the spatial constraints of the nucleus. This packaging has long been appreciated as a passive barrier to DNA access by nuclear factors. The discovery that chromatin regulators play critical roles in virtually all nuclear processes has informed a more nuanced view of chromatin as a dynamic regulatory platform that coordinates access to the genetic material. The smallest unit of chromatin is the nucleosome, a DNA-protein complex composed of ~150 bp of DNA wrapped around an octamer of histone proteins (*Luger et al., 1997*). Nucleosomes can further interact with each other and with other factors to form higher-order structures (*Luger et al., 2012*). Consequently, cells have evolved several sophisticated strategies to regulate chromatin structure at the nucleosome level. These include the covalent modification of histone proteins and DNA, as well as non-covalent changes to the position or composition of nucleosomes at specific genomic loci. Many of the non-covalent transformations, ranging from sliding nucleosomes to the complete disassembly of the histone octamer, are catalyzed by ATP-dependent chromatin remodeling enzymes (*Zhou et al., 2016*). Underscoring their central role in chromatin regulation, remodeling enzymes play essential roles in many processes including transcription, DNA replication, and DNA repair (*Falbo and Shen, 2006*; *Hota and Bruneau, 2016*; *Price and D'Andrea, 2013*). How a relatively small number of remodeler types carry out such diverse regulatory functions remains an area of active research, not least because much

**eLife digest** Every human cell contains nearly two meters of DNA, which is carefully packaged to form a dense structure known as chromatin. The building block of chromatin is the nucleosome, a unit composed of a short section of DNA tightly wound up around a spool-like core of proteins called histones.

The tight structure of the nucleosome prevents the cell from accessing and 'reading' the genes in the packaged DNA, effectively switching off these genes. So the exact placement of nucleosomes helps manage which genes are turned on. Changing the position of the nucleosomes can 'free' the DNA and make genes available to the cell. Enzymes called chromatin remodelers move nucleosomes around – for example, they can make the histone core slide on the DNA strand. However, it is still unclear how these enzymes recognize nucleosomes. Previous research indicates that many proteins bind to nucleosomes by using a surface on the histone proteins called the acidic patch. Could chromatin remodelers also work by interacting with this acidic patch?

To address this further, Gamarra et al. investigate how a chromatin remodeler enzyme known as SNF2h interacts with a nucleosome. By default, SNF2h is inactive because two of its regions called AutoN and NegC act as brakes. The experiments show that the acidic patch helps to bypass this inactivation and switches on SNF2h. Gamarra et al. propose that, when SNF2h docks on to the nucleosome, the patch provides a landing pad for the AutoN and NegC modules; this interaction activates the enzyme, which can then start remodeling the nucleosome. However, another type of chromatin remodeler also uses the patch to interact with nucleosomes but it does not have the AutoN and NegC regions. This suggests that chromatin remodelers work with the acidic patch in different ways. Overall, the findings deepen our understanding of how DNA is packaged in cells, and how this process may go wrong and cause disease.

DOI: https://doi.org/10.7554/eLife.35322.002

remains unknown regarding remodeler mechanisms for substrate recognition and the coupling of that recognition to activity.

Chromatin remodelers are members of the SF2 superfamily of nucleic acid motors, which catalyze noncovalent changes to nucleic acid substrates (*Zhou et al., 2016*). Chromatin remodelers, however, are unique in that they specifically mobilize DNA in the context of the nucleosome, where DNA is tightly bound to histone proteins. Remodelers are further classified into families based on the domain architecture of their ATPase subunit. These families differ in their specific biochemical activities (*Zhou et al., 2016*). Substantial progress in our general understanding of remodeling mechanisms has been made by asking what elements of the nucleosome are important for remodeling in different families. For example, maximal remodeling by ISWI family remodelers, which primarily slide nucleosomes, requires DNA flanking the nucleosome and the N-terminal tail of histone H4 (*Clapier et al., 2001*; *Yang et al., 2006*). Conversely, maximal remodeling by SWI/SNF family remodelers, which carry out the most diverse set of changes to nucleosome structure, does not require these nucleosomal epitopes (*Guyon et al., 1999*). Less is known about how these substrate cues are recognized and mechanistically coupled to remodeling. Some important insights have come from biochemical analyses and structures of remodelers in the absence of nucleosomes, which suggest that in the ground state, chromatin remodelers are held in an inactive conformation by family-specific autoinhibitory motifs (*Clapier and Cairns, 2012*; *Hauk et al., 2010*; *Xia et al., 2016*; *Yan et al., 2016*). Binding to specific nucleosomal epitopes is thought to relieve this autoinhibition via conformational changes in the remodeler, but the details of this process remain unclear.

At the same time, structures of several nucleosome-protein complexes are revealing that many of these proteins interact with a conserved acidic patch formed by histones H2A and H2B on the top surface of the nucleosome (*McGinty and Tan, 2016*). These proteins interact with the acidic patch using an 'arginine anchor' which nestles into a pocket formed by the α1–2 helices of histone H2A (*McGinty and Tan, 2016*). It is therefore plausible that chromatin remodelers also recognize the acidic patch. Indeed recent work has indicated that mutating the acidic patch reduces the activity of ISWI, SWI/SNF and some CHD family remodeling enzymes (*Dann et al., 2017*). To investigate the mechanistic role of the acidic patch in nucleosome remodeling by the ISWI family of enzymes we

used a combination of ensemble and single molecule methods in the context of the human ISWI enzyme, SNF2h. We observe that interactions with the acidic patch activate SNF2h by relieving auto-inhibition mediated by two conserved domains of ISWI enzymes. Our results further suggest that contacts with the acidic patch helps control the distance the nucleosome is moved during transloca-tion events. Finally we find that the acidic patch also stimulates the activity of the INO80 complex. Together, these results highlight the broad and essential role the acidic patch plays in chromatin regulation.

## Results

### The acidic patch of the nucleosome is important for nucleosome sliding by SNF2h

The human ISWI ATPase, SNF2h, preferentially slides nucleosomes toward longer flanking DNA (*Yang et al., 2006*). As a consequence of this activity, SNF2h slides mononucleosomes towards the center of a DNA strand, an activity that can be detected by a native gel mobility assay (*Yang et al., 2006*). Recent work has shown that mutating residues in the nucleosome acidic patch reduces the ability of SNF2h to expose nucleosomal DNA to restriction enzymes (*Dann et al., 2017*). To investi-gate the mechanism by which such mutations affect the nucleosome sliding activity of SNF2h and to determine if these mutations affected SNF2h binding or catalysis, we first used a mutant H2A in which four key residues in the H2A acidic patch are replaced with alanines (E61A, E64A, D90A, and D92A) (*Figure 1A*). We call nucleosomes in which all four of these H2A residues have been mutated to alanines 'APM' (acidic patch mutant) nucleosomes. Using a native gel mobility assay, we mea-sured remodeling rates of wild type (WT) and APM mononucleosomes containing 60 bp of DNA flanking one end of the nucleosome (0/60) (*Figure 1B*). APM nucleosomes are centered substantially slower than WT nucleosomes under conditions where SNF2h is in excess of nucleosomes (*Figure 1B and C*). Since the acidic patch has been shown to be critical for nucleosome binding by several chro-matin proteins, we expected the apparent $K_m$, $K_m^{app}$, to be increased with APM nucleosomes. Inter-estingly however, $K_m^{app}$ is affected substantially less by mutation of the acidic patch than the maximal rate constant for remodeling, $k_{max}$ (*Figure 1D*, 61 nM to 280 nM, corresponding to ~4.5 fold increased $K_m^{app}$, versus 2.5 $min^{-1}$ to 0.012 $min^{-1}$, corresponding to ~200 fold reduced $k_{max}$). These results indicate that the acidic patch plays a larger role in regulating maximal nucleosome slid-ing activity than nucleosome binding. We also measured ATP hydrolysis under conditions where nucleosomes are in excess of SNF2h. At saturating nucleosome concentrations, APM nucleosomes stimulate ATP hydrolysis 4-fold less than WT (*Figure 1G*, *Figure 1—figure supplement 2*). Together, these results imply that the acidic patch plays a critical role in coupling ATP hydrolysis to remodeling after the binding step.

To further investigate the importance of the acidic patch, we asked whether binding by another factor to the acidic patch could compete for nucleosome sliding by SNF2h. We used a peptide derived from the latency associated nuclear antigen (LANA) from Kaposi's sarcoma-associated her-pesvirus that has previously been shown to interact directly with the acidic patch via an arginine anchor (*Barbera et al., 2006*). We carried out these experiments using sub-saturating concentrations of SNF2h. Under these conditions most of the nucleosomes are unbound, allowing direct competi-tion between the LANA peptide and SNF2h for the acidic patch. The presence of the LANA peptide dramatically slows remodeling by SNF2h in a dose-dependent manner (~200 fold reduction in rate constant at 50 μM LANA peptide), while a peptide with a mutant arginine anchor does not have a detectable inhibitory effect (*Figure 1E and F*). We measured a $K_I$ of 1.2 μM for inhibition by the LANA peptide, within 5-fold of its published affinity for the nucleosome (*Figure 1F*) (*Fang et al., 2016*). These results indicate that remodeling by SNF2h requires an interaction with the acidic patch that is mutually exclusive with the binding of acidic patch interacting factors such as the LANA pep-tide. The results are also consistent with recent work showing that the LANA peptide can inhibit the restriction enzyme accessibility activity of the SNF2h containing complex, ACF (*Dann et al., 2017*).

We next investigated if the acidic patch residues act independently or cooperatively by measur-ing the effects of individual mutations. Interestingly, except for residue E64, all the single alanine mutants have comparable defects as the four point mutants combined (*Figure 1—figure*

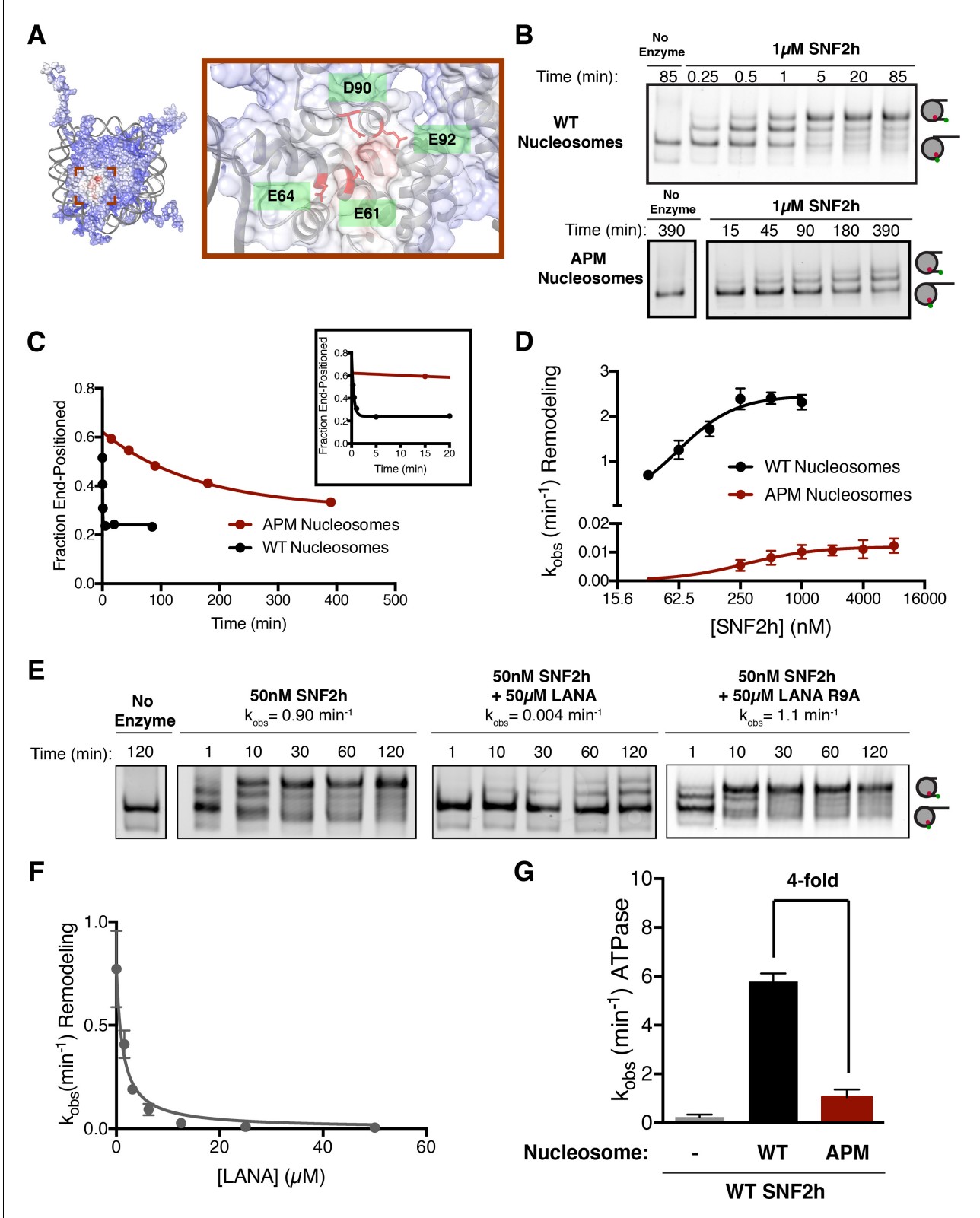

**Figure 1.** The acidic patch is an important epitope for remodeling post-binding. (A) Charge profile of the nucleosome (left) and magnification of the acidic patch region (right) (PDBID: 1KX5, charge profile generated using ABPS and UCSF Chimera [*Pettersen et al., 2004*]). Residues of the acidic patch mutated in this study are shown in red. (B) Example of a Cy5-fluorescent scan of a gel-based remodeling assay with 0/60 WT and acidic patch mutant (APM) nucleosomes (1 μM SNF2h, 20 nM nucleosomes, saturating ATP). (C) Quantification of the gel in B and fits to a single exponential decay.

*Figure 1 continued on next page*

*Figure 1 continued*

Inset is zoomed to show faster time points with WT nucleosomes. (**D**) Gel remodeling rate as a function of enzyme concentration, plotted on a log(2) scale. APM nucleosomes are remodeled substantially slower even at saturating concentrations of enzyme. Data were fit to a cooperative binding model (WT nucleosomes: $K_M^{app}$ = 61 nM, $k_{max}$ = 2.5 min$^{-1}$, h = 1.5; APM nucleosomes: $K_M^{app}$ = 280 nM, $k_{max}$ = 0.012 min$^{-1}$, h = 1.2). (**E**) Remodeling is inhibited by competition for the acidic patch. WT nucleosomes were remodeled with sub-saturating SNF2h and in the presence of KSHV LANA peptide. Remodeling is inhibited by the peptide but not when the arginine anchor is mutated to alanine. (**F**) Inhibition curve with the LANA peptide ($K_I$ = 1.21 µM). Error bars represent standard errors on the mean for three replicates, except for the no-peptide condition in F, which had two replicates. (**G**) ATPase activity of WT SNF2h. Nucleosomes were in excess of SNF2h and at saturating concentrations. APM nucleosomes stimulated ATPase activity 4-fold weaker than WT.

DOI: https://doi.org/10.7554/eLife.35322.003

The following source data and figure supplements are available for figure 1:

**Source data 1.** Values used to obtain plots.
DOI: https://doi.org/10.7554/eLife.35322.009

**Figure supplement 1.** Sequences of DNA constructs used in this work.
DOI: https://doi.org/10.7554/eLife.35322.004

**Figure supplement 2.** Representative ATPase assay fits.
DOI: https://doi.org/10.7554/eLife.35322.005

**Figure supplement 2—source data 1.** Values used to obtain plots.
DOI: https://doi.org/10.7554/eLife.35322.006

**Figure supplement 3.** Maximal remodeling rates for various acidic patch mutations.
DOI: https://doi.org/10.7554/eLife.35322.007

**Figure supplement 3—source data 1.** Values used to obtain plots.
DOI: https://doi.org/10.7554/eLife.35322.008

*supplement 3*). This observation suggests that three of the four acidic patch residues tested here act cooperatively to promote sliding by SNF2h.

## The AutoN and NegC regions of SNF2h cooperate with the acidic patch to enable maximal remodeling

We next investigated which regions of SNF2h might functionally interact with the acidic patch. In principle, these would include (i) regions that directly contact the acidic patch and (ii) regions that do not directly contact the acidic patch, but whose function is energetically coupled to the presence of the acidic patch.

To investigate regions that may directly contact the acidic patch we carried out cross-linking mass spectrometry using the zero-length, carbodiimide based reagent EDC. This method catalyzes the formation of new amide bonds between protein carboxylates, such as the side chains of aspartate and glutamate residues, and amino groups and is therefore well suited to probing electrostatic inter-actors of the acidic patch. Hundreds of high confidence cross-linked residue pairs were identified using this approach (*Supplementary file 1*). To focus on mechanistically meaningful domain-domain interactions, we employed a semi-quantitative mass spectrometry method to compare the extent of SNF2h-nucleosome cross-linking in the presence of different nucleotides (*Figure 2B*). In previous work we have shown that ADP•BeF$_x$ mimics an activated state of the SNF2h-nucleosome complex (*Leonard and Narlikar, 2015*; *Racki et al., 2014*; *Sinha et al., 2017*). We therefore, focused on domain level interactions that were enriched at least two-fold in the presence of ADP•BeF$_x$ relative to ADP.

Cross-links between the H4 tail and RecA lobe 2 of SNF2h are strongly enhanced in the ADP•BeF$_x$ state (*Figure 2B*). This result is consistent with previous work showing that the H4 tail activates ISWI remodelers and promotes a restricted active site conformation in the presence of ADP•BeF$_x$ (*Clapier et al., 2001*; *Hamiche et al., 2001*; *Racki et al., 2014*). We also found that the ADP•BeF$_x$ state promotes specific cross-links between the H2A/H2B acidic patch and lysines in the extended AutoN region, RecA lobe 1, the NegC region, and the DNA binding HAND-SANT-SLIDE (HSS) region of SNF2h (*Figure 2—figure supplement 4*). While cross-links between the acidic patch and the AutoN, HSS and NegC regions are compatible with structural constraints from previous studies, the cross-links with the RecA lobe one are not easily explained. Multiple studies of ISWI remodelers suggest that their RecA lobes bind at a location two DNA helical turns from the nucleosome dyad

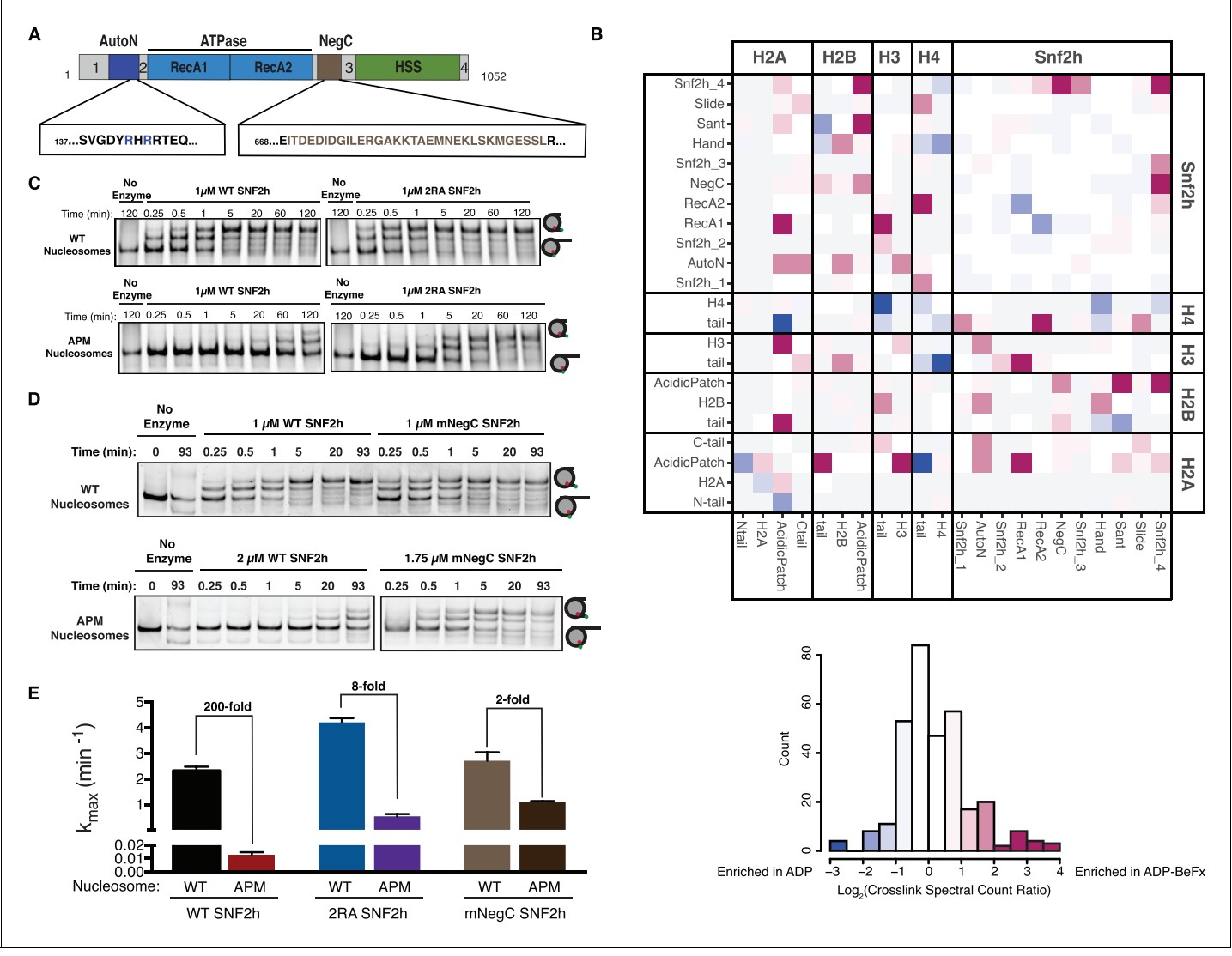

**Figure 2.** Dependence on the nucleosome acidic patch is linked to relief of autoinhibition. (**A**) Domain architecture of SNF2h. The two critical arginines (R142, R144) of AutoN are highlighted in blue, while the NegC region replaced with a flexible GGS linker (*Leonard and Narlikar, 2015*) is highlighted in brown. Intervening sequences with no known domain annotations are numbered as in B. (**B**) Direct domain-domain interactions of SNF2h nucleosomes probed by crosslinking mass spectrometry. SNF2h mononucleosomes were crosslinked with the zero-length reagent EDC in the presence of either the ATP transition state analog ADP•BeF$_x$ or ADP. Residue-residue crosslink data were aggregated into domain-domain level spectral counts. The fold change between ADP•BeF$_x$ and ADP conditions is displayed as the log2 of the spectral count ratio. The data were centered to the median value and the color scale depicts the interactions most enriched in ADP•BeF$_x$ in magenta, and ADP in blue. Light grey tiles indicate no crosslinks observed in either condition. Domains are listed from N- to C- term within each protein. The distribution of fold changes in domain-domain spectral counts is plotted as a histogram below. It should also be noted that due to the structure of the histone octamer and the 2:1 stoichiometry of the SNF2h-Nucleosome complex, we cannot distinguish between intramolecular and intermolecular SNF2h-SNF2h or Histone-Histone crosslinks. (**C** and **D**). Mutation of AutoN or NegC increases remodeling rate with WT and APM nucleosomes. Cy5 fluorescent scans of native gel remodeling assays. (**E**) Saturating remodeling rates ($k_{max}$) for WT and APM nucleosomes with WT, 2RA, and mNegC SNF2h. Mutation of the acidic patch has a substantially lower effect on the $k_{max}$ for 2RA and mNegC SNF2h than for WT SNF2h (8-fold and 2-fold respectively vs. 200-fold). Error bars represent standard errors on the mean for N = 3 replicates.

DOI: https://doi.org/10.7554/eLife.35322.010

The following source data and figure supplements are available for figure 2:

**Source data 1.** Values used to obtain plots.
DOI: https://doi.org/10.7554/eLife.35322.019

**Figure supplement 1.** Mutations to the N-terminus have minimal effects on remodeling.
DOI: https://doi.org/10.7554/eLife.35322.011

*Figure 2 continued on next page*

*Figure 2 continued*

**Figure supplement 1—source data 1.** Values used to obtain plots.

DOI: https://doi.org/10.7554/eLife.35322.012

**Figure supplement 2.** Ensemble FRET remodeling shows similar effects as the gel-based assay.

DOI: https://doi.org/10.7554/eLife.35322.013

**Figure supplement 2—source data 1.** Values used to obtain plots.

DOI: https://doi.org/10.7554/eLife.35322.014

**Figure supplement 3.** Saturation Kinetics Experiments.

DOI: https://doi.org/10.7554/eLife.35322.015

**Figure supplement 3—source data 1.** Values used to obtain plots.

DOI: https://doi.org/10.7554/eLife.35322.016

**Figure supplement 4.** Acidic Patch-SNF2h Crosslinks Unique to the ADP•BeF$_x$ condition.

DOI: https://doi.org/10.7554/eLife.35322.017

**Figure supplement 5.** Comparison of crosslinks in the ADP•BeFx state with and without LANA peptide.

DOI: https://doi.org/10.7554/eLife.35322.018

(SHL ±2), quite far from the H2A/H2B acidic patch (*Dang and Bartholomew, 2007*; *Kagalwala et al., 2004*; *Schwanbeck et al., 2004*; *Zofall et al., 2006*). We hypothesize that the RecA-acidic patch cross-links arise from a population of higher-order SNF2h-nucleosome aggregates in our mass-spec samples and therefore focus below on cross-links to the remaining three regions.

To test the functional significance of cross-links to AutoN, we mutated the lysines that crosslink to the acidic patch in the ADP•BeF$_x$ state. However, none of the mutants significantly altered remodeling by SNF2h (*Figure 2—figure supplement 1*). These results suggest that the lysines in the AutoN region are in proximity to the acidic patch but do not make mechanistically significant interactions. We speculated, however, that other residues in AutoN might make functional interactions with the acidic patch. Previous work has indicated that many nucleosome binding proteins recognize the acidic patch via arginine residues. Near the AutoN lysines that crosslink to the acidic patch resides a key autoinhibitory motif specific to ISWI family remodelers that contains two arginine residues (*Figure 2A*, *Figure 2—figure supplement 4*, R142 and R144). Autoinhibition by these arginine residues is relieved by a basic patch on the H4 tail, a nucleosomal epitope essential for maximal remodeling by ISWI-family remodelers (*Clapier and Cairns, 2012*; *Clapier et al., 2002*). We tested whether R142 and R144 functionally cooperate with the acidic patch by generating a mutated version of SNF2h with the two critical arginines of AutoN mutated to alanine (2RA), and measured the remodeling activity of this mutant. Consistent with previous reports, 2RA SNF2h remodels WT nucleosomes ~ 2 fold faster than WT SNF2h (*Clapier and Cairns, 2012*; *Yan et al., 2016*). However, 2RA SNF2h remodels APM nucleosomes ~ 50 fold faster than WT SNF2h (*Figure 2F*). This corresponds to a ~ 25 fold reduced dependency on the acidic patch for remodeling with 2RA SNF2h (*Figure 2F*). The same trend was also seen by an ensemble FRET remodeling assay (*Figure 2—figure supplement 2*). Together, these data suggest that the acidic patch contributes to relief of autoinhibition by R142 and R144 in AutoN. Since arginine residues are known to mediate this interaction in other systems, binding of either of the two arginines in AutoN to the acidic patch could provide a physical explanation for acidic patch recognition. However, given that neither carbodiimide chemistry nor any other commonly used cross-linking chemistry labels arginine residues, we cannot determine whether R142 and R144 make direct contacts with the acidic patch based on our mass-spectrometry data. We were unable to observe detectable binding between an AutoN peptide containing the 2R residues and the nucleosomal acidic patch through pull down assays (data not shown), suggesting that either these residues do not physically interact with the acidic patch or the surrounding regions of the SNF2h protein are required for stable binding.

To investigate the functional significance of cross-links to the NegC region of SNF2h, we determined the effect of replacing a stretch of 32 residues in NegC with a flexible serine-glycine linker (mNegC). NegC is another autoinhibitory region of SNF2h that imposes flanking DNA length sensitivity on SNF2h by specifically slowing down remodeling of nucleosomes without flanking DNA (*Leonard and Narlikar, 2015*). Consistent with previous work, mNegC SNF2h slides WT 0/60 nucleosomes ~ 1.2 fold faster than WT SNF2h (*Figure 2E and F*) (*Leonard and Narlikar, 2015*). However, mNegC SNF2h slides APM nucleosomes ~ 100 fold faster than WT SNF2h (*Figure 2F*). As

a result, sliding of APM nucleosomes by mNegC SNF2h is only ~2 fold slower than WT nucleosomes. Thus the mNegC mutation almost completely rescues the defect of the acidic patch mutation. These results suggest that residues in NegC also link activation of nucleosome sliding to acidic patch recognition.

The third category of cross-links entailed lysine residues in the HSS regions. Mutants in these lysines greatly reduced remodeling and, in contrast to the 2RA and mNegC SNF2h mutants, did not rescue the defects caused by the acidic patch mutations (data not shown). It is possible that the defects caused by these lysine mutations reflect direct contacts between the HSS residues and the acidic patch. However, because the HSS also contacts flanking DNA, the reduction in sliding rate could also arise from defects in binding DNA.

To better understand which crosslinks are dependent on H2A acidic patch binding, we also performed SNF2h-nucleosome cross-linking reactions with ADP•BeF$_x$ in the presence of the LANA peptide which competes for acidic patch binding (*Figure 1E*). Addition of the LANA peptide substantially reduced crosslinks between the H2A acidic patch and both AutoN and NegC (*Figure 2—figure supplement 5*). In contrast, crosslinks between the HSS and the acidic patch were less strongly affected by LANA addition (*Figure 2—figure supplement 5*). This suggests that HSS positioning near the acidic patch in the ADP•BeF$_x$ state is not strictly dependent on direct binding to the region of the H2A acidic patch contacted by the LANA peptide. As a result, we cannot unambiguously interpret HSS-acidic patch crosslinks as reflecting mechanistically significant interactions. Together, these data suggest that acidic patch recognition is strongly linked to relief of autoinhibition by NegC and AutoN for nucleosome sliding by SNF2h. Furthermore, activation of SNF2h involves conformational changes that bring both of these autoinhibitory domains and the HSS in closer proximity to the acidic patch.

## The acidic patch is required to promote the translocation phase of the SNF2h reaction

To gain additional insight into which steps in the remodeling cycle involve interaction with the acidic patch, we turned to a single-molecule FRET assay (smFRET; *Figure 3A*). This assay is analogous to the ensemble FRET remodeling assay described in *Figure 2—figure supplement 2*, in that the activity of SNF2h in sliding nucleosomes away from DNA ends increases the distance between a donor and acceptor dye pair, and thus decreases the measured FRET efficiency. However, with smFRET we can follow the remodeling of individual, surface-immobilized nucleosomes, and thereby gain insights into the activity of SNF2h reaction steps that are obscured in asynchronous, population-averaged ensemble assays. smFRET has previously been used to study remodeling by several ISWI family members, as well as by remodeling complexes from the SWI/SNF and INO80 families (*Blosser et al., 2009*; *Deindl et al., 2013*; *Harada et al., 2016*; *Hwang et al., 2014*; *Zhou et al., 2018*). A key insight revealed by these smFRET studies is that ISWI family remodelers do not slide nucleosomes in a continuous manner, such that FRET decreases continuously over time, but rather in a series of alternating phases: a 'pause' phase, in which FRET (and therefore nucleosome position) remain constant, and a 'translocation' phase, in which the nucleosome is moved relative to the DNA end. These repeating pause phases, which so far appear to be specific to ISWI, are essential to our understanding of the mechanism of action of ISWI remodelers, because the overall remodeling rates observed in ensemble assays are dominated by the durations of the pause phases, not the translocation rate itself. Moreover, substrate cues such as the H4 tail and flanking DNA have been shown to be sensed in the pause phase, not the translocation phase, for ISWI family remodelers (*Hwang et al., 2014*). That is, shorter flanking DNA or mutation of the H4 tail increases the durations of the pauses, thereby decreasing the overall remodeling rate, while having no effect on the actual sliding rate of the nucleosome. These results suggest a separation of regulation and activity in ISWI remodelers: translocation events are regularly interrupted by pauses that allow for the periodic interrogation of substrate cues. This pausing behavior also explains the ability of ISWI remodelers such as ACF to keep nucleosomes centered: if the nucleosome is translocated off-center, the interruption of this translocation by a new pause can trigger translocation in the opposite direction, restoring the nucleosome to a centered position.

As shown in *Figure 3B*, we find that SNF2h alone, like the ACF complex and the yeast ISWI enzymes, shares this alternating pause and translocation behavior at the single-molecule level. We first wished to ascertain whether the acidic patch, like other nucleosomal epitopes recognized by

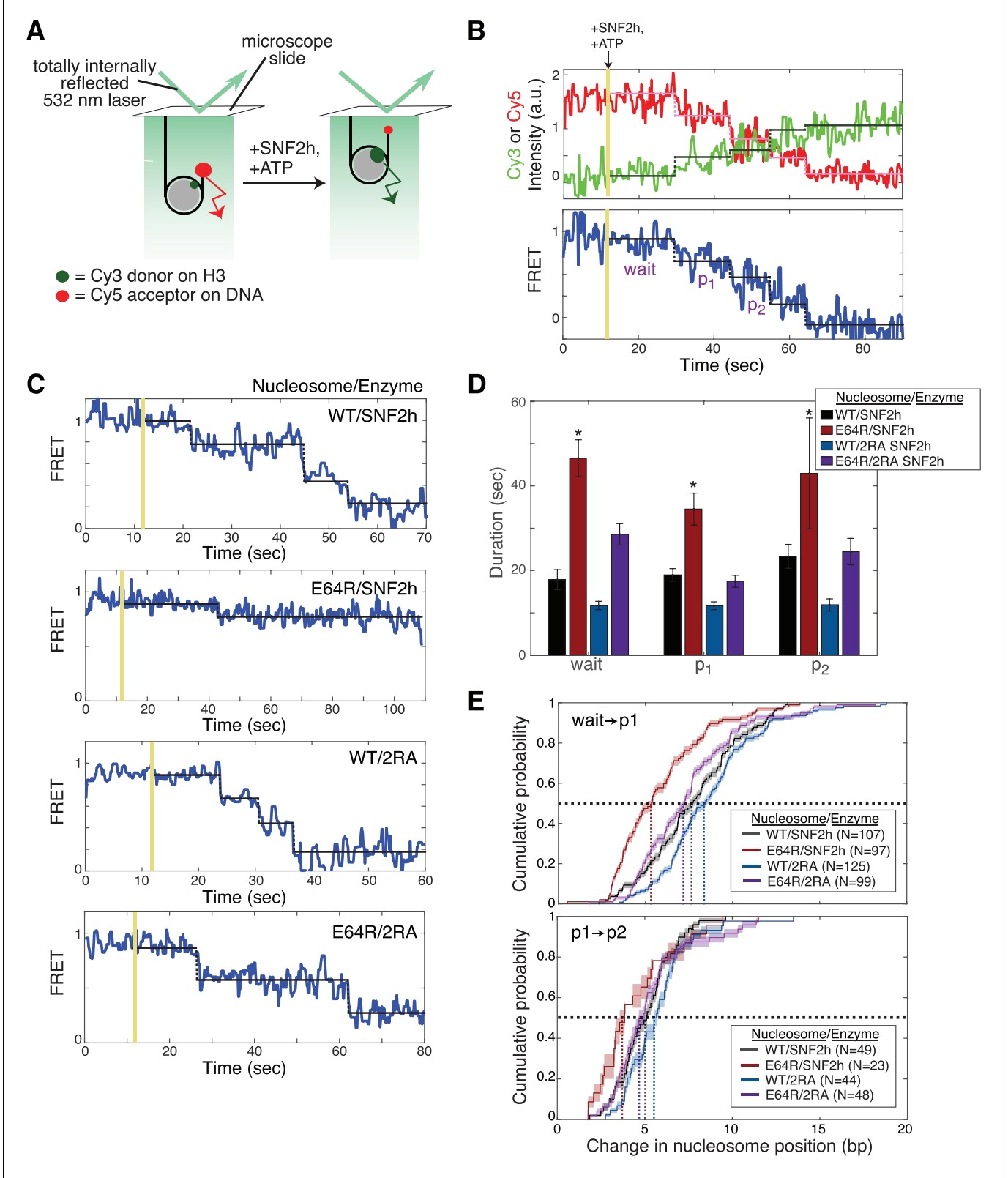

**Figure 3.** The acidic patch interacts antagonistically with AutoN to promote pause exit and persistent translocation. (**A**) Schematic of the smFRET setup. Nucleosomes labeled on histone H3 with a Cy3 donor dye and on one end of the DNA with a Cy5 acceptor dye are immobilized on the surface of a slide and imaged with a prism-based TIRF microscope. The end-positioned nucleosomes used here have an initial high FRET efficiency (see also *Figure 3—figure supplement 2*). As remodeling proceeds and the nucleosome is moved away from the DNA end, the FRET efficiency is reduced. (**B**)
*Figure 3 continued on next page*

*Figure 3 continued*

Example time-course of remodeling of a single surface-attached, WT nucleosome in the presence of saturating SNF2h and ATP (400 nM and 1 mM, respectively). Vertical yellow line indicates addition of enzyme and ATP; horizontal pink, green, and black lines are the output of an HMM fit used to quantify pause durations and locations (see Methods). Note the alternating 'pause' and 'translocation' phases of the remodeling reaction; in keeping with the ISWI literature, we call the first pause the 'wait' pause, the second pause 'p1', the third 'p2', etc. Intensity and FRET data here and in C have been smoothed with a 0.95 s median filter for visualization only. (C) Example time-courses of remodeling of, from top to bottom, WT nucleosomes by SNF2h, E64R nucleosomes by SNF2h, WT nucleosomes by 2RA SNF2h, and E64R nucleosomes by 2RA SNF2h. Saturating enzyme (400 nM enzyme with WT nucleosomes; 2 µM enzyme with E64R nucleosomes) and ATP (1 mM) were used in all cases. Additional examples are shown in *Figure 3—figure supplement 2*. (D) Quantification of the first three pause durations. *Indicates a lower limit; as shown in *Figure 3—figure supplement 2B*, remodeling of E64R nucleosomes by SNF2h is too slow to capture by smFRET, due to the competition between photobleaching of the dyes and remodeling by SNF2h. Errors were bootstrapped (see Materials and methods). (E) Empirical cumulative distribution functions (cdfs) of the change in nucleosome position during the first translocation event (top panel) and the second translocation event (bottom panel) for different combinations of nucleosome constructs and enzymes. Roughly 50% of the initial translocation events by SNF2h on WT nucleosomes move the nucleosome 7 bp or fewer (black dashed lines); in contrast, nearly 80% of initial translocation events by SNF2h on E64R nucleosomes move the nucleosome 7 bp or fewer. Similarly, during the second translocation event, SNF2h moves WT nucleosomes roughly 5 bp or fewer 50% of the time, whereas again nearly 80% of second translocation events for SNF2h with E64R nucleosomes result in step sizes 5 bp or fewer. Shaded areas represent bootstrapped error estimates (see Materials and methods). See also Figure 3—figure supplement 6 for additional representations of these data and a further discussion of step sizes.
DOI: https://doi.org/10.7554/eLife.35322.020

The following source data and figure supplements are available for figure 3:

**Source data 1.** Values used to obtain plots.
DOI: https://doi.org/10.7554/eLife.35322.024
**Figure supplement 1.** smFRET controls.
DOI: https://doi.org/10.7554/eLife.35322.021
**Figure supplement 2.** Additional example traces of SNF2h or SNF2h/2RA remodeling single nucleosomes, plotted as in *Figure 3C*.
DOI: https://doi.org/10.7554/eLife.35322.022
**Figure supplement 3.** The step size of WT SNF2h with WT nucleosomes is comparable to step sizes previously described for ISWI family remodelers.
DOI: https://doi.org/10.7554/eLife.35322.023

ISWI remodelers, affects the regulatory pause phase. Because the rate of remodeling of APM nucleosomes by SNF2h is significantly slower (on the order of hours) than the rate of dye photobleaching (on the order of minutes; *Figure 3—figure supplement 1B*), we assembled nucleosomes with a single point mutation in the acidic patch (E64R nucleosomes). We used the E64R mutation rather than the E64A mutation because the defect caused by this mutation was better rescued by the 2RA mutation in SNF2h than the defect in E64A (*Figure 1—figure supplement 3*). The single-point mutation (E64R) is less deleterious than mutating all 4 acidic residues (APM), and SNF2h remodels E64R nucleosomes ~ 40 fold more slowly than WT nucleosomes as opposed to 200-fold more slowly with APM nucleosomes (*Figure 1—figure supplement 3*). However, this remodeling rate is still very slow relative to the timescales typically measured by smFRET, which posed two additional challenges: an increase in the number of noise events (dye blinking, intensity fluctuations, etc) per remodeling trajectory, and an increase in the amount of data to be analyzed. These challenges were addressed through the use of custom in-house smFRET analysis software ('Traces', https://github.com/stephlj/Traces) (*Zhou et al., 2018*; *Johnson et al., 2018*; copy archived at https://github.com/elifesciences-publications/Traces) to streamline the analysis of large data sets, and the adaptation of a computationally fast, versatile, open-source hidden Markov model (HMM) library called pyhsmm to quantify the durations of the pauses (see Materials and methods).

As shown in the example trajectories in *Figure 3C*, remodeling of E64R nucleosomes by SNF2h proceeds through the same alternating pause and translocation phases as does remodeling of WT nucleosomes. However, the pauses are noticeably longer with E64R nucleosomes, by at least a factor of 2 (*Figure 3D*), indicating that the acidic patch epitope, like other substrate cues, is indeed sensed during the regulatory pause phase. We note that remodeling of E64R nucleosomes in ensemble assays is significantly slower than the photobleaching rate (*Figure 3—figure supplement 1B*), so that by smFRET we only detect the fraction of remodeling events that are faster than photobleaching. As a result, the remodeling rate obtained for the E64R nucleosomes by smFRET represents an upper bound on the true remodeling rate (i.e. the E64R nucleosomes appear to remodel faster by smFRET than by ensemble assays (*Figure 3—figure supplement 1B*)).

Given the ability of the 2RA mutation of the AutoN motif to partially rescue remodeling defects in APM nucleosomes in ensemble assays (*Figure 2C*), we next asked whether this rescue is due to a restoration of wild-type pause durations. In agreement with our ensemble results (*Figure 2C*), 2RA SNF2h remodels WT nucleosomes slightly faster than SNF2h by reducing pause durations (*Figure 3D*). The reduction is minor (~1.3 fold), consistent with a previous smFRET study of the effect of the 2RA mutation on pause durations for the ACF complex (*Hwang et al., 2014*). Furthermore, 2RA rescues the deleterious effect of the E64R acidic patch mutation, by nearly restoring wild-type pause durations (*Figure 3D*). The effects of the E64R nucleosomal mutation and the 2RA SNF2h mutation on pause durations therefore mirror the effects on overall remodeling rates of APM nucleosomes: 2RA remodels E64R nucleosomes nearly as fast as SNF2h remodels WT nucleosomes, but not as fast as 2RA remodels WT nucleosomes (*Figure 1—figure supplement 3*). These results are consistent with a model in which relief of autoinhibition of the AutoN motif of SNF2h through direct or indirect interactions with the acidic patch enable pause exit (that is, promote the translocation phase).

The example traces in *Figure 3C* suggest an additional defect in remodeling of E64R nucleosomes: a reduction in the distance the nucleosome is moved during each translocation event (which we called the step size). Note that after two translocation events, WT nucleosomes with SNF2h or 2RA are moved from ~0.95 FRET to ~0.4 FRET (*Figure 3C* top and second from bottom), whereas after two translocation events the nucleosome in the E64R/SNF2h example trace has moved from ~0.95 FRET to ~0.75 FRET. Step size, like pause duration, plays an important role in regulating ISWI remodeler activity: since the pause durations dominate the overall remodeling rate, a smaller step size, which means more pauses per unit distance that the nucleosome is translocated, will mean a significant reduction in the overall remodeling rate (as observed in ensemble assays).

Step sizes can be quantified by converting the change in FRET between subsequent pauses to a change in the number of base pairs of DNA between the Cy5-labeled DNA end and the edge of the nucleosome. We accomplish this conversion by means of a calibration curve, described previously (*Zhou et al., 2018*). Like other ISWI family remodelers, SNF2h moves WT nucleosomes with an initial large step (~8 bp) followed by a smaller (~5 bp) step (*Figure 3E*, *Figure 3—figure supplement 3*; [*Blosser et al., 2009*; *Deindl et al., 2013*]). However, SNF2h moves E64R nucleosomes a shorter distance in each translocation phase, as indicated by the leftward shift of the cumulative distributions in the red curves of *Figure 3E*, relative to the black curves (on average, E64R nucleosomes move about 6 bp in the first translocation and about 4 bp in the second (*Figure 3—figure supplement 3*)). Mutation of AutoN has little effect on the step size in the context of WT nucleosomes, but largely restores the step size in the context of E64R nucleosomes to that of SNF2h with WT nucleosomes (*Figure 3E*, magenta and blue curves). Given that ISWI family remodelers have been shown to translocate a nucleosome in elementary steps of 1–2 bp (*Deindl et al., 2013*), our results suggest that with E64R nucleosomes, SNF2h takes fewer of these elementary steps in succession during the translocation phase, before entering a new pause phase.

The major contributions to the remodeling defects observed with SNF2h and APM nucleosomes can therefore be attributed to two effects: first, an increase in pause duration, and second, a decrease in the distance travelled per translocation event, meaning that there are more (and longer) pauses in APM remodeling reactions per unit distance that the nucleosome is moved. Both of these effects are rescued by the 2RA mutation of the AutoN motif of SNF2h. We propose that the acidic patch is important for relieving autoinhibition by AutoN and thereby promoting exit from the pause phase. Further, we propose that the acidic patch is also involved in keeping AutoN out of the active site until the nucleosome has been translocated to the full extent (~8 bp initially,~5 bp subsequently) and a new pause phase is entered (Figure 5).

## The acidic patch is used by both ISWI and INO80 complexes

The ISWI ATPase forms complexes with several accessory proteins that regulate its localization and activity (*He et al., 2006*; *Oppikofer et al., 2017*; *Tsukiyama et al., 1995*; *Varga-Weisz et al., 1997*). ACF is one of the best studied of these complexes. ACF is a heterodimer of the ISWI ATPase subunit (SNF2h in humans) and the accessory subunit Acf1, and is implicated in gene repression, DNA replication, and DNA repair (*Collins et al., 2002*; *Fyodorov et al., 2004*; *Lan et al., 2010*). Biochemically, human ACF has the same core activity as SNF2h, but displays greater nucleosome affinity, enhanced sliding rates, and better kinetic discrimination of flanking DNA length (*He et al.,*

*2006; Yang et al., 2006*). Despite these similarities, recent evidence suggests that ACF has some mechanistic differences from SNF2h on its own. For instance, in the context of ACF, AutoN regulates flanking DNA length sensing through interaction of an Acf1-specific domain called WAC (*Hwang et al., 2014*). SNF2h alone has no comparable domain. Furthermore, recent work has suggested that mutating the nucleosomal acidic patch causes a smaller defect in remodeling by ACF compared to SNF2h (*Dann et al., 2017*). Given this difference, we asked whether ACF requires the acidic patch for remodeling beyond binding. At saturating concentrations of ACF, where binding does not contribute to the overall remodeling rate, ACF slides APM nucleosomes 10-fold more slowly than WT (*Figure 4A*), indicating that ACF also uses the acidic patch in a step after binding. However, consistent with previous work, ACF is less dependent on the acidic patch compared to SNF2h alone (*Dann et al., 2017*).

We next asked whether ISWI-family complexes uniquely use the acidic patch, or whether it is also used by other chromatin remodeling enzymes. Recently, it was shown that while some CHD family remodelers slide nucleosomes largely independent of the acidic patch, others are dependent on it (*Dann et al., 2017; Levendosky et al., 2016*). It has been further shown that SWI/SNF family enzymes also require the acidic patch for maximal activity (*Dann et al., 2017*). These observations raise the possibility that the acidic patch is a feature used by most remodeling enzymes. To address this issue we determined if the acidic patch is required by yeast INO80, which is in a distinct family from the CHD, ISWI and SWI/SNF families. INO80, like ACF, slides nucleosomes preferentially in the direction of longer flanking DNA and can also create evenly spaced nucleosome arrays (*Udugama et al., 2011*). This sliding activity is thought to be important for positioning the +1 nucleosome at transcription start sites (*Krietenstein et al., 2016*). Interestingly, we find that INO80 slides APM nucleosomes ~ 200 fold more slowly than WT nucleosomes at saturating concentrations (*Figure 4B*). This result indicates that INO80 also uses the acidic patch post-binding, but is more dependent on the acidic patch than the ACF complex.

## Discussion

In this study, we investigate the role of the highly conserved H2A acidic patch in chromatin remodeling by ISWI enzymes. We find that the acidic patch is used post-binding in order to activate remodeling by both INO80 and ISWI family remodelers. Furthermore, using a combination of ensemble and single molecule methods, we show that the acidic patch is used by SNF2h to relieve autoinhibition by the conserved AutoN and NegC motifs. Below we explore the mechanistic and regulatory implications of these results.

ATP-dependent chromatin remodeling enzymes carry out specialized reactions on a complex substrate. Understanding how recognition of this substrate is coupled to activity can provide a means to understanding common principles underlying ATP-dependent chromatin remodeling mechanisms. Owing to decades of study, ISWI enzymes provide a useful model system to address this question. On the basis of crosslinking and footprinting studies, the ATPase domain of ISWI enzymes is thought to bind and translocate DNA two helical turns from the nucleosomal dyad (SHL ± 2) (*Dang and Bartholomew, 2007; Kagalwala et al., 2004; Schwanbeck et al., 2004; Zofall et al., 2006*). Work from several groups has also shown that for ISWI remodelers, recognition of both the H4 tail and flanking DNA enhances remodeling activity post-binding (*Clapier et al., 2001; Hamiche et al., 2001; Yang et al., 2006*). While the mechanisms by which these nucleosome cues activate remodeling are not well understood, the ISWI domains that recognize these cues are known. The C-terminal HAND-SANT-SLIDE (HSS) domain mediates flanking DNA recognition, while the H4 tail appears to be directly recognized by the second RecA lobe within the ATPase domain (*Dang and Bartholomew, 2007; Yan et al., 2016*). The acidic patch resides on a surface near SHL ± 6, far from where the ATPase domain engages the nucleosome. How then might the acidic patch be recognized and used by ISWI remodelers?

Our results suggest that two known autoinhibitory regions of ISWI enzymes, the AutoN region and the NegC region, functionally interact with the acidic patch, because mutating these regions dramatically reduces the dependence of SNF2h on the acidic patch for sliding. This suggests that a large role of the acidic patch is to relieve autoinhibition by both AutoN and NegC. While we do not have evidence for a direct interaction between the acidic patch and the two arginines in AutoN, our cross-linking mass spectrometry data suggests that activation of the enzyme places residues within

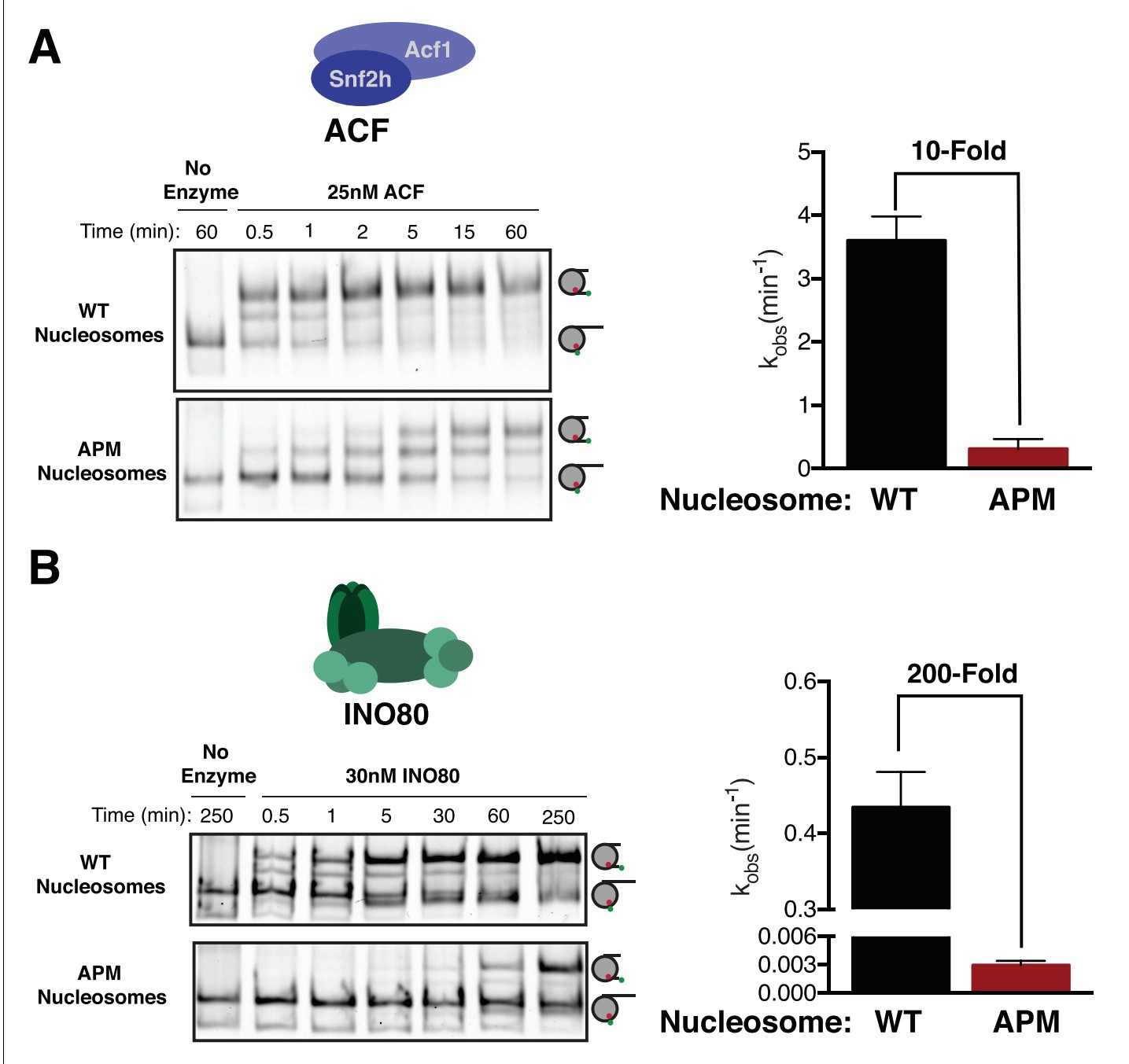

**Figure 4.** The acidic patch is used by ACF and INO80. (**A**) Gel remodeling assay with human ACF and 0/60 nucleosomes. Saturating concentrations of enzyme and ATP were used. (**B**) Gel remodeling with the yeast INO80 complex and 0/60 nucleosomes. Reactions were performed with saturating enzyme and ATP. Error bars represent standard error of the mean for N = 3 replicates.

DOI: https://doi.org/10.7554/eLife.35322.025

The following source data is available for figure 4:

**Source data 1.** Values used to obtain plots.
DOI: https://doi.org/10.7554/eLife.35322.026

both AutoN and NegC near this location. Our smFRET work here and previous smFRET work suggests that the AutoN region inhibits the transition from the pause phase to the translocation phase (*Hwang et al., 2014*). It has been shown that flanking DNA and the H4 tail are both sensed in the pause phase (*Hwang et al., 2014*). Further, previous ensemble work has suggested that the NegC region inhibits the transition between a flanking DNA sensing state of SNF2h and a translocation competent state of SNF2h (*Leonard and Narlikar, 2015*). We therefore propose that the acidic patch helps promote the translocation competent state of SNF2h by providing an alternative binding site for NegC and AutoN (*Figure 5*).

In addition to an increase in pause durations with the acidic patch mutations, the amount of DNA translocated within a translocation phase is reduced compared to WT nucleosomes. We hypothesize that translocation is interrupted by premature reversion of the enzyme to the autoinhibited state in the absence of stabilizing interactions with the acidic patch (*Figure 5*). As a result, more pauses are encountered per distance translocated. Our results lead to a model in which the acidic patch provides a binding surface for NegC and AutoN that sequesters these regions from inhibiting SNF2h (*Figure 5*). Combined with previous work, our results underscore how the strong coupling of relief of autoinhibition to recognition of two conserved nucleosome cues (the H4 tail and the acidic patch) make this motor exquisitely specific for its complex substrate.

The acidic patch increases $K_m^{app}$ for SNF2h by ~5 fold, suggesting that interactions with the acidic patch also stabilize SNF2h binding. Thus the acidic patch appears to be used in at least two distinct steps of the SNF2h remodeling reaction. What residues in SNF2h might be interacting with the acidic patch in these two steps? In contrast to the rescuing effects of AutoN mutations on maximal activity, AutoN mutations additively increase $K_m^{app}$ beyond solely mutating the acidic patch (*Figure 2—figure supplement 3*). This suggests that AutoN and the acidic patch do not cooperate in the ground state. It is thus possible that different SNF2h domains contact the acidic patch in different steps of the remodeling reaction. The different set of cross-links observed between the acidic patch and SNF2h regions in the ADP vs. ADP•BeF$_x$ state are consistent with such a possibility.

Based on previous work, we have hypothesized that SNF2h undergoes a large conformational change prior to adopting a translocation competent state that repositions the C-terminus from binding flanking DNA towards binding the nucleosome core (*Leonard and Narlikar, 2015*). Our crosslinking-MS data provides additional insights into the structural rearrangements that accompany enzyme activation. We find that crosslinks between the H2A/H2B acidic patch and the SANT domain increase in the ADP•BeF$_x$-bound state compared to the ADP-bound state (*Figure 2B*), consistent with the HSS binding to the nucleosome core. Interestingly, these crosslinks are not substantially reduced by LANA peptide binding, suggesting that the location of the HSS in the ADP•BeF$_x$-bound state is not strongly dependent on direct contacts with the residues contacted by the LANA peptide

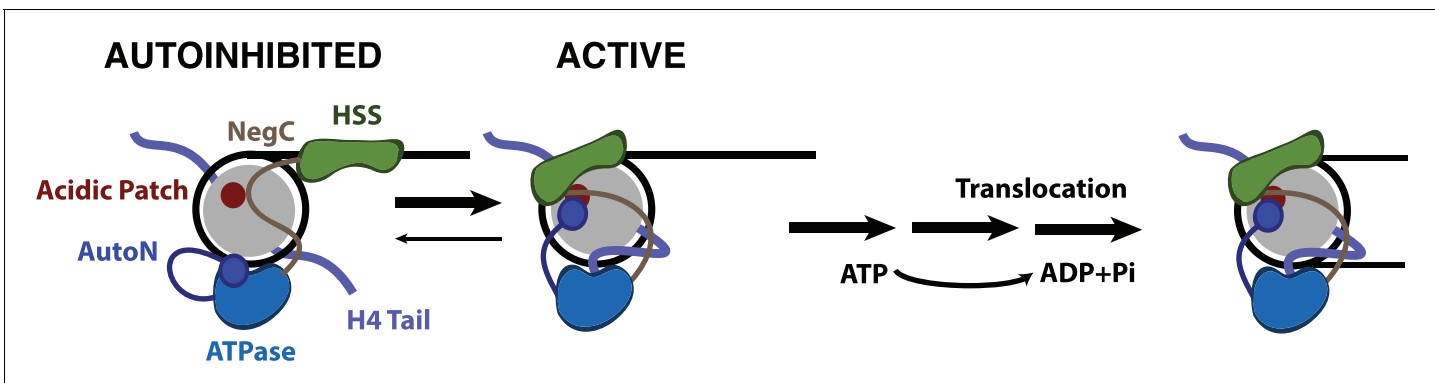

**Figure 5.** Model for nucleosome remodeling by SNF2h. After binding the nucleosome, SNF2h is in equilibrium between an active and autoinhibited state. In the autoinhibited state AutoN and NegC hold the remodeler in an inactive state. The active state is promoted by AutoN and NegC binding near the acidic patch and by H4 tail binding the ATPase domain resulting in conformational changes that bring HSS in close proximity to the acidic patch. From this active state, SNF2h can translocate DNA around the octamer. Although SNF2h remodels nucleosomes as a dimer at saturating enzyme concentrations (*Blosser et al., 2009*; *Racki et al., 2009*), in this model we display only one of the protomers for simplicity.
DOI: https://doi.org/10.7554/eLife.35322.027

(*Figure 2—figure supplement 5*). This result raises the possibility that the SANT domain and LANA peptide occupy adjacent regions on the nucleosome core. The SANT domain may then cross-link to the acidic patch when these sites are transiently exposed due to dynamics in LANA peptide binding. In contrast, NegC and AutoN crosslinks to the acidic patch are substantially reduced by the LANA peptide, suggesting LANA binding directly or indirectly displaces these regions from the nucleosome. Overall our results suggest that the key accessory regions of SNF2h namely, HSS, NegC and AutoN, are all positioned near the acidic patch in the activated state. In agreement with this observation, crosslinks between the C-terminus of SNF2h and NegC increase in the ADP•BeF$_x$ state (*Figure 2B*). The close positioning of multiple SNF2h accessory domains near the acidic patch raises the possibility that contacts between accessory domains may play a role in promoting the translocation-competent state. While substantial future work would be needed to test this possibility, it is analogous to recent observations with the yeast CHD1 remodeling motor that contacts between its N-terminus and the C-terminal DNA binding domain regulate the sliding reaction (*Sundaramoorthy et al., 2017*).

Most chromatin remodeling ATPases also form large multi-subunit complexes, which regulate the activity and specificity of the remodeling reaction. Here we find that the human ISWI complex, ACF, requires the acidic patch for maximal activity but shows a ~ 20 fold smaller defect upon mutation of the acidic patch than observed with SNF2h alone. This result is qualitatively consistent with recent studies showing that the acidic patch has a smaller role in the activity of ACF vs. SNF2h (*Dann et al., 2017*). Our results here provide a mechanistic framework for understanding these recent observations. In particular, our results suggest that the accessory protein Acf1 alters the mechanism of relief of autoinhibition by the acidic patch in a manner that makes the reaction less dependent on the acidic patch, perhaps by providing an alternative binding partner for the AutoN and NegC domains. Such a domain would be analogous to the WAC motif of Acf1, which provides an alternative binding partner for the H4 tail (*Hwang et al., 2014*). In contrast to ACF, we find that the yeast INO80 complex, a large multisubunit complex, is as dependent on the acidic patch for nucleosome sliding as the isolated SNF2h ATPase. Importantly, INO80 family remodelers are insensitive to the presence of the H4 tail and no AutoN-like or NegC like motif has been identified in the ATPase subunit of INO80 (*Udugama et al., 2011*). The acidic patch must then activate INO80 through a mechanism distinct from ISWI complexes. Determining how the acidic patch is used by INO80 ATPases, and what roles the ATPase and accessory subunits play in this mechanism, are important areas of future study.

Combined with previous results (*Dann et al., 2017*), our results suggest that several families of chromatin remodelers require the acidic patch for remodeling. However, it is possible that this surface is not a universal requirement for chromatin remodeling, as yeast CHD1 can remodel nucleosomes largely independent of the acidic patch (*Levendosky et al., 2016*). CHD1 instead uses an unidentified aspect of the histone H2A/H2B dimer to promote remodeling (*Levendosky et al., 2016*). Yeast CHD1 and ISWI family remodelers have been thought to share a common remodeling mechanism, as these families share biochemical activities, like nucleosome sliding and spacing, and substrate cues required for maximal remodeling activity, such as the H4 tail and flanking DNA (*Ferreira et al., 2007*; *Stockdale et al., 2006*). However, the enormous difference in dependence on the acidic patch between yeast CHD1 and ISWI enzymes raise the possibility that these families remodel nucleosomes through distinct mechanisms (*Levendosky et al., 2016*). Importantly, while yeast CHD1 shares domains with ISWI remodelers such as the SANT-SLIDE domains at the C-terminus and a version of the NegC region, called the C-terminal bridge (*Hauk et al., 2010*; *Ryan et al., 2011*), CHD family remodelers do not appear to possess an AutoN motif. Instead, remodelers like CHD1 have an N-terminal double chromodomain which has an analogous role as an autoinhibitiory domain that is relieved by H4 tail binding (*Hauk et al., 2010*).

At a primary level, the requirement of the acidic patch provides a powerful means for remodelers to sense and respond to chromatin structure and nucleosome content. Thus, nucleosomes lacking histone H2A-H2B dimers or containing a modified acidic patch through histone variants or covalent modifications may be recognized and remodeled differently than canonical nucleosomes by different remodelers (*Dann et al., 2017*). Consistent with this possibility, nucleosomes containing histone H2Az, which have an extended acidic patch, are remodeled ~2 fold faster by ISWI complexes than canonical nucleosomes (*Goldman et al., 2010*). Analogously, recent work has shown that INO80 preferentially slides H2AZ nucleosomes over H2A nucleosomes (*Brahma et al., 2017*). Finally, given the growing list of factors that recognize the acidic patch, it is likely that remodelers and other

chromatin binding proteins compete for access to the acidic patch. Indeed, binding by the LANA peptide to the acidic patch competes for remodeling by SNF2h. Sensitivity to the acidic patch could be a general mechanism to regulate the outcome of chromatin remodeling at loci where remodelers and other factors are jostling for access to their chromatin substrates.

# Materials and methods

## Expression and purification of chromatin remodeling enzymes

SNF2h was purified from *E. coli* as described previously with minor modifications (*Leonard and Narlikar, 2015*). DNA was precipitated following cell lysis by addition of 5% w/v polyethylenimine (P3143, Sigma-Aldrich, St. Louis, MO) pH 7.9 dropwise to a final concentration of 0.1% and clarified by centrifugation. Following cobalt affinity purification, the 6xHis tag was cleaved overnight with TEV protease and dialysed into SEC Buffer. TEV-cleaved SNF2h was then purified by anion exchange chromatography using a HiTrap Q column and size exclusion chromatography (GE Life Sciences, Pittsburgh, MA). SNF2h concentration was determined by SDS-PAGE with BSA protein standards and staining with SYPRO Red (Thermo Fisher, Waltham, MA).

Human ACF was expressed and purified recombinantly from Sf9 insect cells by FLAG immunoaffinity purification as described previously with minor modifications (*Aalfs et al., 2001*). SNF2h-FLAG and Acf1 were expressed separately via infection with baculovirus. Nuclear extracts from each construct were generated and mixed together at a 10:1 Acf1:SNF2h-FLAG volume ratio to ensure full assembly of the complex. This mixture was then bound to FLAG M2-affinity resin (Sigma-Aldrich, St. Louis, MO), washed with increasing KCl concentrations, and eluted with buffer with 100 mM KCl and 1 mg/mL FLAG Peptide. ACF concentration was determined by SDS-PAGE with BSA standards and based on the intensity of the Acf1 band.

INO80 was purified by FLAG immunoprecipitation based on previously published methods (*Shen, 2004*; *Zhou et al., 2018*). Briefly, *S. cerevisae* with endogenously flag-tagged INO80 was grown in YEPD at 30°C to saturation. Cells were pelleted by centrifugation for 10 min at 5000 rpm, resuspended with buffer H0.3 (25 mM HEPES, pH 7.5, 1 mM EDTA, pH 8.0, 10% glycerol, 0.02% NP-40, 0.3 M KCl), and pelleted again. Pelleted cells were then extruded through a 60 mL syringe into liquid nitrogen to create 'noodles'. Cell 'noodles' were then lysed using a Tissue Lyser II (Qiagen, Hilden Germany) cooled in liquid nitrogen. Frozen lysate powder was resuspended in equal volume of H0.3 and spun in an SW28 rotor for 2 hr at 25,000 rpm at 4°C. Clarified lysate was mixed with equal volume buffer H0.3 and applied to FLAG M2-affinity resin (1 mL bead slurry per 40 mL of cleared lysate) equilibrated with H0.3 and incubated for 3 hr at 4°C. Resin was washed with 3 × 50 mL buffer H0.5 (H0.3 buffer except with 0.5 M KCl) followed by 3 × 10 mL washes with buffer H0.1 (0.1M KCl) and eluted with H0.1 supplemented with 1 mg/mL FLAG peptide. Eluate was concentrated, aliquoted, flash frozen in liquid nitrogen, and stored at −80°C. INO80 concentration was determined by SDS-PAGE with BSA standards, based on the intensity of the Ino80-flag band.

## Nucleosome labeling and reconstitution

Recombinant *Xenopus laevis* histones were expressed and purified from *E. coli* as previously described (*Luger et al., 1999*). Histone H2A E61A, E64A, D90A, D92A expression plasmid was a generous gift from the Tan lab at Penn State. Purified histone H2A E64R was provided by the Wolberger lab. Histone octamer was reconstituted as previously described (*Luger et al., 1999*; *Zhou and Narlikar, 2016*), except for smFRET nucleosomes where a 2:1 unlabeled:labeled H3 mixture was used during octamer assembly to generate nucleosomes with mostly one H3 or neither H3 labeled. Histone H3 with a cysteine introduced at position 33 was labeled with either Cy3 (for smFRET) or Cy5 (for ensemble assays) prior to histone octamer assembly via cysteine-maleimide chemistry. Cy3-labeled (for ensemble assays) and Cyanine 5 SE-labeled and biotinylated DNAs (for smFRET) were generated by PCR with HPLC-purified, labeled primers (Cy5 primers: TriLink Biotechnologies, San Diego, CA; Cy3 and biotinylated primers: IDT, Coralville, IA) and purified by PAGE. The strong, synthetic 601 nucleosome positioning sequence (*Lowary and Widom, 1998*) was used to assemble all nucleosomes in this study, with an arbitrary sequence for DNA flanking the 601 positioning sequence (*Figure 1—figure supplement 1*). These DNAs were assembled with either wild-

type or APM octamers by salt gradient dialysis and purified by glycerol gradient centrifugation (*Zhou and Narlikar, 2016*).

## Native gel remodeling assay

All remodeling reactions were performed under single turnover conditions (enzyme in excess of nucleosomes). Reactions with SNF2h were performed at 20°C with 20 nM nucleosomes, 12.5 mM HEPES pH 7.5, 2 mM Tris pH 7.5, 70 mM KCl, 5 mM ATP•MgCl$_2$, 3 mM MgCl$_2$ , 0.02% NP40, and ~3%(v/v) glycerol. Reactions with ACF and INO80 were performed as above at 30°C and with minor changes in buffer composition (ACF: 10 nM nucleosomes, 12.5 mM HEPES pH 7.9, 2 mM Tris pH 7.5, 60 mM KCl, 2 mM ATP•MgCl$_2$, 3 mM MgCl$_2$, 0.02% NP40, 0.3 mg/mL FLAG peptide, and ~9% glycerol; INO80: 10 nM nucleosomes, 40 mM Tris pH 7.5, 60 mM KCl, 2 mM ATP•MgCl$_2$, 1.1 mM MgCl$_2$, 0.02% NP40, 0.5 mg/mL FLAG peptide, and 1% glycerol). Reactions were started with addition of enzyme and time points were quenched with excess ADP and plasmid DNA. Time points were then resolved by native PAGE (6% acrylamide, 0.5XTBE) and scanned on a Typhoon variable mode imager (GE Life Sciences, Pittsburgh, PA) by scanning for fluorescent labels. Gels were then quantified by densitometry using ImageJ. The fraction of nucleosomes end-positioned (i.e. unremodeled) at a given time point was determined by the ratio of fast-migrating nucleosomes to the total nucleosome intensity. This was fit to a single exponential decay using Prism 6 (GraphPad, La Jolla, CA) (*Equation 1*),

$$y = (y_0 - p)e^{-k_{obs}t} + p \tag{1}$$

where $y_0$ is the initial fraction end-positioned, $k_{obs}$ is the observed rate constant, and $p$ is the fraction end-positioned at plateau. Reactions in a given concentration series were fit constrained to a common $y_0$ and $p$. Concentration series were fit to a cooperative binding model (*Equation 2*),

$$k_{obs} = k_{max}\frac{X}{(K_m^{app})^h + X^h} \tag{2}$$

where $X$ is the concentration of SNF2h, $h$ is the hill coefficient, $K_m^{app}$ is the apparent $K_m$, and $k_{max}$ is the saturating rate constant. Competition assays were performed as described above with varying concentrations of LANA peptide and fit to a single exponential decay. This was then fit to a simple competition binding model (*Equation 3*),

$$k_{obs} = \frac{k_0}{1 + \frac{X}{K_I}} \tag{3}$$

where $k_0$ is the rate constant without peptide, $X$ is the concentration of peptide, and $K_I$ is the inhibition constant.

## ATPase assay

ATPase reactions were performed under multiple turnover conditions (nucleosomes in excess of enzymes). Reactions were performed with 12.5 nM SNF2h, 12.5 mM HEPES pH 7.5, 2 mM Tris pH 7.5, 70 mM KCl, 7.5 μM ATP•MgCl$_2$, 3 mM MgCl$_2$, 0.02% NP40,~3%(v/v) glycerol, and trace amounts of γ-$^{32}$P-ATP. Reactions were started with addition of enzyme, and 2.5 μL time points were quenched with an equal volume of 50 mM Tris pH 7.5, 3% SDS, and 100 mM EDTA. Inorganic phosphate was resolved from ATP on a PEI-cellulose TLC plate (Select Scientific) with 0.5 M LiCl/1M formic acid mobile phase. Plates were dried, exposed to a phosphorscreen overnight, and scanned on a Typhoon variable mode imager. Rate constants were determined by fitting a line through the first 10% of inorganic phosphate generated using Prism.

## Ensemble FRET remodeling assay

Ensemble FRET remodeling assays were performed under the same conditions as gel remodeling assays. Reactions were initiated by addition of enzyme and then measured in a K2 fluorometer (ISS) equipped with a 550 nm short pass excitation filter and a 535 nm long pass emission filter. Reactions were excited at 515 nm and emission was measured at 665 nm. The resulting curves were fit to a two-phase exponential decay (*Equation 4*),

$$y = (p + (y_0 - p)(f_{fast} e^{-k_{fast} t}) + (1 - f_{fast}) e^{-k_{slow} t}) \tag{4}$$

where $f_{fast}$ is the fraction in the fast phase and $k_{fast}$ and $k_{slow}$ are the apparent rate constants of the fast and slow phase respectively.

## Crosslinking mass spectrometry

Crosslinking mass spectrometry samples were prepared by incubating 72 μg of mononucleosomes without flanking DNA at 9 μM final concentration with 24 μM SNF2h in buffer containing either ADP or ADP•BeF$_x$ (15 mM HEPES pH 7.5, 140 mM KCl, 0.5 mM ADP•Mg, 0.5 mM MgCl$_2$,±0.5 mM BeFx 1:5 BeCl$_2$:NaF) for 10 min at 30°C. The crosslinking reaction with the LANA peptide was performed the same as the ADP•BeFx condition with 30 μM peptide added. The samples were then reacted with 1 mM EDC and 20 μM N-hydroxysulfosuccinimide (added as a 10x stock in water) for 60 min at room temperature. Crosslinking reactions were then quenched by adding 10 mM Tris pH 7.5 and samples were acetone precipitated and washed once with cold acetone. The pellet was resuspended in 8M Urea, 5 mM TCEP, 100 mM ammonium bicarbonate and heated at 56°C for 25 min, followed by alkylation with 10 mM iodoacetamide for 40 min at room temperature. The sample was diluted 5-fold with 100 mM ammonium bicarbonate and digested with 1:25 trypsin for 4 hr at 37°C followed by addition of a second aliquot of trypsin and overnight digestion.

Crosslinked peptides were desalted using 100 μl OMIX C18 tips (Agilent), fractionated by size-exclusion chromatography (SEC), and analyzed by LC-MS similarly to a previously described method (*Zhou et al., 2017*). Briefly, trypsin digests were acidified to 0.2% TFA, desalted, and run over a Superdex Peptide PC 3.2/300 s column (GE Healthcare). SEC fractions eluting between 0.9 ml and 1.4 ml were dried and resuspended in 0.1% formic acid for LC-MS. Each fraction was separated over a 15 cm x 75 μm ID PepMap C18 column (Thermo) using a NanoAcquity UPLC system (Waters) and analyzed by a Fusion Lumos mass spectrometer (Thermo). Precursor ions were measured from 375 to 1500 m/z in the Orbitrap analyzer (resolution: 120,000; AGC: 4.0e5). Ions charged 3+ to 8+ were isolated in the quadrupole (selection window: 1.6 m/z units; dynamic exclusion window: 30 s; MIPS Peptide filter enabled), fragmented by HCD (Normalized Collision Energy: 28%) and measured in the Orbitrap (resolution: 30,000; AGC; 5.0e4). The cycle time was set to 3 s.

Peaklists were generated using PAVA (UCSF) and searched for crosslinked peptides with Protein Prospector 5.19.22 (*Trnka et al., 2014*) against a target database containing human SNF2h plus the four core histone sequences from *X. laevis* concatenated with a decoy database containing 10 randomized copies of each target sequence (total database size: 55 sequences). Loss of the initiator methionine and carbamidomethylation of cysteine. Methionine oxidation, peptide N-terminal glutamine to pyroglutamate formation, acetylation at the protein N-terminus, and mis-annotation of the monoisotopic peak (1 Da neutral loss) were treated as variable modifications. EDC was designated as a heterobifunctional crosslinking reagent with specificity of aspartate, glutamate, and the protein C-terminus on one side and lysine and the protein N-terminus on the other with a bridge mass corresponding to loss of H$_2$O. A mass modification range of 400–5000 Da was specified on these residues and 85 product ion peaks from the peaklist were used in the search. Precursor and product ion tolerances were 8 and 25 ppm respectively.

Crosslinked spectral matches (CSMs) were initially classified as in (*Zhou et al., 2017*). The dataset was then aggregated into unique crosslinked residue-pair level data with a corresponding spectral count value. Due to the prevalence of multiple, closely spaced Asp and Glu residues in a typical tryptic peptide, site-localization of EDC crosslinks is more challenging than with homobifunctional lysine-directed reagents. To address this, when the site-localization was judged to be ambiguous, all possible residue-pairs were kept with an annotation noting the ambiguity. When calculating spectral counts, fractional spectral counts were assigned to these ambiguous site localizations so that a given spectrum was awarded exactly one spectral count. For instance, a product ion spectrum matching equally well to both K91.H4-D65.H2B or K91.H4-E68.H2B contributes 0.5 towards the spectral counts of each residue-pair. Decoy CSMs were retained throughout this aggregation and spectral counting process. A linear SVM model, built on five features of the Protein Prospector search output (score difference, % of product ion signals matched, precursor charge, rank of peptide 1, and rank of peptide 2) was constructed to sort crosslinked residue pairs into decoy and target classes. Crosslinked residue-pairs with an SVM score greater than 1, score difference greater than 5, and at least

one spectral count are reported. The final residue-pair level data set is reported at specificity of 99.7% corresponding to 0.05% FDR. The number of unique crosslinks in the ADP condition was 974, while 1470 crosslinks were unique to the ADP•BeF$_x$ condition and 707 crosslinks were common to both conditions.

To determine which protein domains are involved in SNF2h mediated nucleosome sliding, residue level crosslink spectral counts were aggregated into domain level counts. Each domain pair was assigned a minimum spectral count of 1 to avoid dividing by zero, and the log2 ratio of spectral counts for each domain pair in the ADP•BeFx condition to the ADP condition were calculated. Domain pairs with a Log2 ratio of exactly 0 were treated as NA values. The remaining data were normalized such that the median value was set at 0. Hence, most domain-domain interactions were assumed to not change substantially between conditions.

Histone protein sequences were taken from *Xenopus Laevis* Uniprot Entries (without the initiator methionine) and domains were defined as follows: H2A N-term tail: 1–16, H2A: 17–43, H2A Acidic Patch: 44–100, H2A C-term tail: 101–129; H2B tail: 1–34, H2B: 35–99, H2B Acidic Patch: 100–122; H3 tail: 1–44, H3: 45–135; H4 tail: 25–102.

The sequence of SNF2h was identical to the Human entry in Uniprot (O60264) with an additional two amino acids at the N-terminus to match the construct used. All residue numbers are therefore shifted from the Uniprot entry by two aa. SNF2h domains were defined as follows: Snf2h1: 1–83, AutoN: 84–160, Snf2h2: 161–183, RecA1: 184–402, RecA2: 403–641, NegC: 642–703, Snf2h4: 704–736, HAND: 737–839, SANT: 840–894, SLIDE: 895–1013, Snf2h5: 1014–1054.

Annotated Mass Spectra are available using MS-Viewer at: http://msviewer.ucsf.edu/prospector/cgi-bin/msform.cgi?form=msviewer

The ADP data set is accessed with search key: 2x0kr2kzq1

The ADP•BeFx data set is accessed with search key: fjamygr8pl

The ADP•BeFx in the presence of LANA peptide is accessed with search key: c5o2mcxwum.

Raw mass spectrometry data is available in the MassIVE repository at UCSD with accession key: MSV000082136

## Single molecule FRET measurements

smFRET experiments were performed as previously described in (*Zhou et al., 2018*) with modifications to the reaction buffers as described below.

### Sample preparation and imaging

Briefly, as in (*Zhou et al., 2018*), quartz slides were PEGylated and then incubated with neutravidin (A2666, ThermoFisher Scientific, Waltham, MA) to mediate attachment of biotinylated nucleosomes. After removal of unbound neutravidin, biotinylated nucleosomes at 12.5 pM in a modified Wash Buffer (12 mM HEPES-KOH, pH 7.5 at 22°C, 60 mM KCl, 1.4 mM MgCl$_2$, 10% glycerol, 0.1 mM EDTA, 0.02% Igepal, 1% [w/v] glucose, and 0.1 mg/mL acetylated BSA) were added and incubated for 10 min. Unbound nucleosomes were removed by washing with Wash Buffer. All incubations and experiments were performed at 20°C. Nucleosomes were imaged on a custom-built prism-based TIRF setup.

Immediately prior to data acquisition, the sample chamber was flushed with a modified imaging buffer (53 mM HEPES-KOH, pH 7.5 at 22°C, 9.1 mM Tris-acetate, pH 7.5 at 22°C [contributed by the Trolox], 63 mM KCl, 1.41 mM MgCl$_2$, 10% glycerol, 0.1 mM EDTA, 0.02% Igepal, 1% [w/v] glucose, 0.1 mg/mL acetylated BSA, 2 mM Trolox [Sigma 238813, made as an 11 mM stock in Tris-acetate, pH'd to 7.5 with 1 M NaOH, and stored at 4°C], 0.03 mM β-mercaptoethanol, 2 U/μL catalase [=0.2 mg/mL; Sigma E3289], and 0.08 U/μL glucose oxidase [0.8 mg/mL, Sigma G2133; made with the catalase as a 100x stock in SPB, and stored at 4°C for not more than one week]). Images were collected using Micro-Manager (www.micro-manager.org, San Francisco, CA) (*Edelstein et al., 2010*) at 7.4 Hz, with an exposure time of 100 ms. To start each reaction, saturating enzyme (400 nM for WT nucleosomes; 2 μM for E64R nucleosomes) and saturating ATP (1 mM) in 300 μL imaging buffer were added via an automated syringe pump (J-KEM Scientific, St. Louis, MO).

## Data analysis

The number of remodeling events per smFRET experiment was roughly an order of magnitude lower with E64R nucleosomes compared to WT nucleosomes, necessitating significantly larger data sets for the E64R nucleosomes compared to WT. To streamline the data analysis process with these data sets, as well as to improve the overall quality of the data, we made use of the custom in-house software we have developed for smFRET image analysis, called Traces, available for download at https://github.com/stephlj/Traces (*Zhou et al., 2018*; *Johnson et al., 2018*; copy archived at https://github.com/elifesciences-publications/Traces). In addition, the long pauses exhibited by SNF2h remodeling E64R nucleosomes are subject to an increased number of artifacts, such as dye blinking or slight fluctuations in the noise, which can complicate quantification of smFRET trajectories. We therefore used the python-based HMM library pyhsmm (https://github.com/mattjj/pyhsmm), which we adapted for the analysis of smFRET data as part of the Traces package, to quantify pause durations. This particular HMM package, which fits a discrete state HMM to each trajectory generated by Traces, enables a reduction in the likelihood of the HMM identifying artifacts as real transitions, and also reduces analysis time.

As described in *Figure 3—figure supplement 1*, and consistent with previous smFRET studies with nucleosomes (*Blosser et al., 2009*; *Deindl et al., 2013*; *Hwang et al., 2014*), we observe two predominant clusters of FRET values, at 0.57 and 0.95 FRET, in the absence of remodeler. These FRET states correspond to two of the four populations of nucleosomes that result from mixing unlabeled H3 with Cy3-labeled H3 during octamer formation: some nucleosomes will have a Cy3 label on the H3 proximal to the Cy5-labeled DNA end, resulting in the higher FRET state, and some will have a Cy3 label on the H3 distal to the Cy5-labeled DNA end, resulting in the mid-FRET state. There will also be a population of nucleosomes with both H3 histones unlabeled, which show no FRET; and a population with both copies of H3 labeled, which are distinguishable by two-step photobleaching of the Cy3 dye, and are excluded from all analyses (i.e. both calibration curve data (*Zhou et al., 2018*) and remodeling data). We also excluded any trajectories to which pyhsmm fit an initial FRET value lower than 0.775 FRET, since nucleosomes with distally labeled H3's do not provide as great a dynamic range for monitoring nucleosome remodeling. We excluded any part of a trajectory including and subsequent to backtracking events (where the nucleosome was moved away from the center, that is, where FRET increased instead of decreasing). Each data set (e.g., E64R/SNF2h) consists of at least 100 trajectories collated from at least 3 (typically 5 to 7) different experiments. All errors were bootstrapped over trajectories (that is, for each data set, the $\geq$100 trajectories were resampled with replacement, and reported values, such as the means of each pause duration, were recalculated for each bootstrapped sample). Reported errors are standard deviations of the bootstrapped values. A similar procedure was used to obtain the errors on the cdfs in *Figure 3E* and *Figure 3—figure supplement 3*; the shaded regions represent ±the standard deviation of the set of bootstrapped cdfs.

## Acknowledgements

We thank Matthew Johnson for help in developing Traces, adapting his physmm code to smFRET data analysis, and the statistical analysis of smFRET data. We thank Song Tan for the Histone H2A E61A, E64A, D90A, D92A expression plasmid, and Greg Bowman, Robert Levendosky, and Cynthia Wolberger for purified Histone H2A E64R and H2A D90R/E92A. We thank Sebastian Deindl for providing an implementation of the Kerssemakers step-finding algorithm (*Kerssemakers et al., 2006*) in Matlab. We also thank Coral Zhou for INO80 purification, Julia Tretyakova for histone purification, Serena Sanuli for help generating some of the nucleosome substrates, and all members of the Narlikar Lab for helpful discussions. This work was supported by a grant from the NIH to GJN (R01GM073767), an NSF predoctoral fellowship and a UCSF discovery fellowship to NG, a Leukemia and Lymphoma Society Career Development Program Fellow award to SJ, and a grant from the Adelson Medical Research Foundation to ALB. The Thermo Scientific Fusion Lumos was funded by the UCSF Program for Breakthrough Biomedical Research (PBBR).

# Additional information

## Competing interests

Geeta J Narlikar: Reviewing editor, *eLife*. The other authors declare that no competing interests exist.

## Funding

| Funder | Grant reference number | Author |
|---|---|---|
| National Science Foundation | Predoctoral Fellowship | Nathan Gamarra |
| University of California, San Francisco | Discovery Fellowship | Nathan Gamarra |
| Leukemia and Lymphoma Society | Career Development Program Fellow Award | Stephanie L Johnson |
| Dr. Miriam and Sheldon G. Adelson Medical Research Foundation | | Alma L Burlingame |
| University of California, San Francisco | Program for Breakthrough Biomedical Research (PBBR) | Alma L Burlingame |
| National Institute of General Medical Sciences | R01GM073767 | Geeta J Narlikar |

The funders had no role in study design, data collection and interpretation, or the decision to submit the work for publication.

## Author contributions

Nathan Gamarra, Conceptualization, Resources, Data curation, Formal analysis, Validation, Investigation, Visualization, Methodology, Writing—original draft, Writing—review and editing, Conceived of and conducted the biochemistry experiments and relevant data analysis, Wrote this manuscript; Stephanie L Johnson, Data curation, Software, Formal analysis, Validation, Investigation, Visualization, Methodology, Writing—original draft, Writing—review and editing, Conceived of and conducted the single molecule experiments and relevant data analysis, Helped write this manuscript; Michael J Trnka, Data curation, Software, Formal analysis, Validation, Investigation, Visualization, Methodology, Writing—review and editing, Designed the crosslinking experiments, Analyzed the mass spectrometry data, Developed analytical methods; Alma L Burlingame, Supervision, Funding acquisition, Project administration, Supervised the mass spectrometry data collection; Geeta J Narlikar, Conceptualization, Supervision, Funding acquisition, Methodology, Writing—original draft, Project administration, Writing—review and editing, Supervised the overall study and helped write this manuscript

## Author ORCIDs

Nathan Gamarra (iD) https://orcid.org/0000-0002-2430-8662
Michael J Trnka (iD) https://orcid.org/0000-0002-8808-5146
Geeta J Narlikar (iD) https://orcid.org/0000-0002-1920-0147

## Decision letter and Author response

Decision letter https://doi.org/10.7554/eLife.35322.043
Author response https://doi.org/10.7554/eLife.35322.044

# Additional files

## Supplementary files

• Supplementary file 1. Crosslinked residue pairs identified from EDC treatment of SNF2h-nucleosomes in the presence of: ADP, ADP-BeFx, or ADP-BeFx and LANA peptide.
DOI: https://doi.org/10.7554/eLife.35322.028

• Transparent reporting form

DOI: https://doi.org/10.7554/eLife.35322.029

## Data availability

Relevant source data is provided in the main and supplemental figures. Crosslinked residue pair identification along with number of spectral counts per identification are reported in Supplementary file 1, as well as in a web resource with links to annotated product ion spectra (see Experimental Methods). Raw mass spectrometry files are available on the Massive server (UCSD). Code used for the analysis of smFRET data can be found at the following link, which is also found in the main text. https://github.com/stephlj/Traces

The following datasets were generated:

| Author(s) | Year | Dataset title | Dataset URL | Database, license, and accessibility information |
|---|---|---|---|---|
| Trnka MJ | 2018 | Annotated product ion peaklists - ADP crosslinked dataset | http://msviewer.ucsf. edu/prospector/cgi-bin/ msform.cgi?form= msviewer | Publicly available at the UCSF MS-Viewer (search key 2x0 kr2kzq1) |
| Trnka MJ | 2018 | Annotated product ion peaklists - ADP•BeFx crosslinked dataset | http://msviewer.ucsf. edu/prospector/cgi-bin/ msform.cgi?form= msviewer | Publicly available at the UCSF MS-Viewer (search key fjamygr8pl) |
| Trnka MJ | 2018 | Unprocessed Mass Spectrometry Files | https://massive.ucsd. edu/ProteoSAFe/data-set.jsp?task=4b2c8ce69f-b44e89890b7617728d19-e6 | Publicly available at MassIVE (identifier MSV000082136) |
| Trnka MJ | 2018 | Annotated product ion peaklists - ADP•BeFx with LANA peptide crosslinked dataset | http://msviewer.ucsf. edu/prospector/cgi-bin/ msform.cgi?form= msviewer | Publicly available at the UCSF MS-Viewer (search key c5o2mcxwum) |

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
