## [Decision Letter]

[Editors’ note: a previous version of this study was rejected after peer review, but the authors submitted for reconsideration. The first decision letter after peer review is shown below.]

Thank you for submitting your work entitled "An activating role for the nucleosome acidic patch in ATP-dependent chromatin remodeling" for consideration by *eLife*. Your article has been reviewed by three peer reviewers,, one of whom is a member of our Board of Reviewing Editors and the evaluation has been overseen by a Senior Editor. The following individuals involved in review of your submission have agreed to reveal their identity: Tom Owen-Hughes (Reviewer #3).

Our decision has been reached after consultation between the reviewers. Based on these discussions and the individual reviews below, we regret to inform you that your work will not be considered further for publication in *eLife*.

As you will see when reading the reviews appended below there was considerable enthusiasm amongst the reviewers for various aspects of the manuscript. However, the lack of evidence of a direct interaction between the acidic patch and AutoN dampened our enthusiasm as this would appear to be essential data to support the proposed model. There were also issues with regards to the measurements of the "step sizes" in the reaction as to how those relate to what is in the published literature. As it is unclear how long it would take to acquire the necessary data, the reviewers concluded that the current manuscript should be rejected, rather than caught in what might be a considerable delay. We would welcome a resubmission should additional data become available. Please feel free to communicate with us should you desire any clarification.

Reviewer 1

This is an exciting manuscript that takes powerful biochemical and biophysical approaches to analyze details of nucleosome remodeling by the Snf2h (Iswi) remodeling ATPase. The results implicate a central role of the acidic patch on the H2A H2B dimer as an important epitope in receiving auto-inhibition of remodeling activity. This effect is mediated by increased pause and decreased steps in nucleosome translocations. The authors go on to show that the ACF complex and Ino80 are also dependent on the acidic patch for remodeling. Some points that need to be addressed.

1) This paper needs to include some biochemical characterization of the APM nucleosomes compared to WT. Are they as stable for example at higher salt, etc. While the location of the mutations suggests they should not affect nucleosome stability, this can't be assumed. Moreover, the fact that Ino80 remodeling is also affected by the mutations without having similar Auto inhibitory domains suggest that the effect of the mutations is on the nucleosome substrate rather than through the remodelers.

2) The assumption that the single point mutant is simply a weaker version of the 4-point mutants in the acidic patch may not be true. It may act completely differently. This assumption would be stronger if additional single point mutants were examined.

3) The mechanism would be strengthened if the authors can demonstrate a direct interaction between AutoN and the acidic patch and that this is reduced in the mutants. Simple peptide pulldown assays with Auto N compared to the H4 tails would suffice.

4) The authors need to make the case that studying hSn2h in the absence of its partner proteins is warranted. The fact that ACF (of which hSf2h is a subunit) is much less dependent on the acidic patch than hSn2h alone could be construed as suggesting these are artifacts of studying hSnf2h out of its normal context.

Reviewer 2

The main findings of this work are that the ISWI and INO80 remodelers are sensitive to the acidic patch on histone H2A. The ISWI ATPase subunit, SNF2h, is more extensively studied, where it is shown that nucleosome sliding is dramatically reduced with a quadruple acidic patch mutant. A nice complementary experiment is the competition with the LANA peptide, which is known to bind to the acidic patch, and therefore serves as a confirmation that this region is important for nucleosome sliding by ISWI. By using saturating amounts of remodeler, the authors demonstrate that the defects in sliding are not solely due to binding (though nucleosome interactions are decreased with the mutants), and therefore the defect is catalytic in nature. The authors find that a common mutant of the autoinhibitory "auto-N" motif (2RA) can rescue the remodeling defects, and they hypothesize a direct connection between basic region of the auto-N and acidic patch of H2A.

1) While I agree with the general conclusions, the single molecule experiments presented in Figure 3 are harder to interpret. The authors propose that the E64R mutant increases the pause durations (which would be similar to what was observed for H4∆tail and short DNA linker nucleosomes in smFRET experiments; see Hwang et al., 2014). The author report that nucleosomes are shifted in steps, as expected for ISWI, and also the intriguing possibility that the E64R mutant alters the step size. A significant concern of mine, however, is that the example traces showing steps (e.g. Figure 3B) were rather noisy, and I would like to see more examples of the steps (as a supplement). Also, the HMM step sizes should agree with step sizes that one can obtain from the step finding algorithm published by Kerssemakers et al., (2006). If the authors assemble the steps into histograms based on FRET values at each step, it should be apparent what the preferential steps are and whether they change with AP nucleosomes. Such analysis would be more informative and appropriate than the cumulative probability plots shown in Figure 3E.

2) On a related point, in the Discussion section, the description of sliding the E64R mutant is confusing with regard to increased pausing frequencies. The authors describe an apparent change in the stepsize in nucleosome sliding, which may or may not result from increased frequency of pausing, but no data has been presented for pausing frequency (distinct from pausing duration). Figure 3E is discussed with regard to the distances translocated, not on frequencies of pausing, so this point is unclear.

3) Based on the core thesis of their work, that the acidic patch relieves inhibition by auto-N and therefore accelerates overall sliding, I think it would be worthwhile to investigate sliding behavior of H2AZ nucleosomes, which are known to have a more extensive acidic patch region. The Kingston lab (Goldman et al., 2010) showed that ISWI is more active with H2AZ nucleosomes. They found that the "additional acidic residues" of H2AZ were required for the higher sliding rate, but somewhat unexpectedly, adding just the additional acidic residues of H2AZ to normal H2A did not appear to alter sliding. While such experiments may not lend new insight for ISWI, H2AZ is a natural substrate for INO80 and therefore it would be worthwhile to compare sliding rates for H2A, H2AZ, and H2AZ-NK (acidic patch=normal H2A).

4) Figure 3—figure supplement 3A: The authors state that 2-dye nucleosomes were excluded from analysis, so that the FRET differences from distal and proximal dyes could be properly accounted for. This is fine, but it is unclear how both dyes together affect the resulting initial FRET. Was there some mechanism for ensuring that traces used for the remodeling experiments were also from single-dye nucleosomes?

5) Figure 3—figure supplement 5: why are the heights of the peaks from the two differently positioned dyes so different. Shouldn't these have similar densities? The 12/78 seems somewhat odd with the closer dye yielding a much lower peak than the more distal dye. Can the authors comment on this?

6) The authors state in the Discussion section "…acidic patch mutations increase both the durations and frequencies of pauses." The authors only present the E64R mutant in smFRET experiments, and so should either present data for other mutants or adjust this sentence accordingly.

Reviewer 3

In this manuscript it is shown that mutation of residues that comprise the considered acidic patch on the nucleosome affect the ability of some, chromatin remodeling enzymes to act on nucleosomes bearing these nucleosomes. This is interesting as this region of the nucleosome has previously been shown to act as a binding interface for other types of nucleosome binding proteins. However, the acidic patch is not universally required for remodeling enzymes as the enzyme Chd1 has previously been shown to act on nucleosomes where this interface is altered. Also, when SNF2h is present as part of the ACF complex the effect of the AP region is reduced 20-fold.

The functional significance of this interaction is investigated further by studying the rate at which a more active mutant of the SNF2h enzyme (AutoN) remodels nucleosomes. This mutant enzyme is found to be less sensitive to alterations to the acidic patch on the nucleosome. As a result, a model is proposed in which AutoN interacts with the acidic patch in the active state, but switches back to an inhibitory conformation during cycles of enzyme activity. Support for this is gained from single molecule studies showing that pausing is increased by mutation of the AP surface. Although the AutoN mutation does not affect pausing on normal nucleosomes, it does act to restore normal enzyme action on AP mutant nucleosomes.

It's known that the AutoN loop interacts with ATPase lobe 2 and that the histone H4 competes for engagement with this region (Yan et al. 2016). Here it is proposed that the acidic patch region acts as an alternative location via which both AutoN and the H4 tail can interact. A major weakness of the manuscript is that no evidence is presented to directly support direct association of these regions. Ideally, dynamic association of AutoN and the H4 tail with the AP region would be directly measured during the reaction cycle. The authors have established a single molecule FRET system that could be used to make this type of measurement, it would none the less be a substantial amount of new work to include this type of work.

Some form of evidence should be obtained to indicate that AutoN and the AP region interact directly. There are a number of different approaches that could be used for this including FRET, X-link MS or pull down. This would provide important evidence supporting the model presented in Figure 5.

The single molecule FRET measurements appear to indicate different lengths of DNA transit over the nucleosome, and that there may be differences from previously published data. This is potentially very interesting. However, the noise in the individual traces appears to be high. The differences in the movements and the errors associated with the measurements should be established rigorously. The authors should also discuss whether the differences in DNA movements are likely to involve differences in the rate at which elementary steps are generated, or the time during which SNF2H is committed to single base steps.

[Editors’ note: what now follows is the decision letter after the authors submitted for further consideration.]

Thank you for resubmitting your work entitled "The nucleosomal acidic patch relieves auto-inhibition by the ISWI remodeler SNF2h" for further consideration at *eLife*. Your revised article has been favorably evaluated by John Kuriyan (Senior editor), a Reviewing editor (Jerry Workman), and three reviewers.

The manuscript has been improved but there are some remaining issues that need to be addressed before acceptance, as outlined below:

1) The EDC cross-linking/mass spec experiments provided new and interesting information regarding how Snf2h binds to the nucleosome. Some of those data, however, appeared to be somewhat selectively interpreted, biased toward the authors' model. While cross-linking of the AutoN domain was observed with the acidic patch, cross-links were also observed for other regions of Snf2h, such as the HAND/SANT/SLIDE (HSS) domain and RecA domains. In addition, a significant number of intramolecular cross-links were observed from the AutoN to other parts of Snf2h. Cross-links closest to 2RA are to other parts of Snf2h rather than the acidic patch. The authors should discuss the potential significance of these interactions. It would be helpful if the authors could devise a figure to more precisely indicate locations of cross-links with respect to 2RA and NegC.

2) There were many fewer cross-links in the ADP compared to the ADP-BeF_x_ complex, which makes it unclear whether the ratios shown in Figure 2B are the most appropriate way to present nucleotide-dependent differences. Were those data normalized? If not, could a set of cross-links not expected to change in response to nucleotide (e.g. Snf2h intradomain, or histone-histone) be used as an internal standard for normalizing the ADP and ADP-BeF_x_ data?

3)The authors should consider additional experiments to determine whether the HSS domain might be the element that directly interacts with the acidic patch. One experiment would be to test whether the 2RA mutant of the AutoN can rescue the Lys-to-Ala mutants in the HSS that the authors' found disrupt activity. A rescue would signify that AutoN bypass is downstream of HSS regulation. Alternatively, the HSS mutants that fail to slide nucleosomes could be compared to wt in EDC cross-linking/mass spec experiments. Selective loss of other HSS cross-links with the acidic patch would support a direct interaction. These experiments are not a requirement for publication, but if feasible on a short time scale could provide additional valuable information.

---

## [Author Response]

[Editors’ note: the author responses to the first round of peer review follow.]

Reviewer 1This is an exciting manuscript that takes powerful biochemical and biophysical approaches to analyze details of nucleosome remodeling by the Snf2h (Iswi) remodeling ATPase. The results implicate a central role of the acidic patch on the H2A H2B dimer as an important epitope in receiving auto-inhibition of remodeling activity. This effect is mediated by increased pause and decreased steps in nucleosome translocations. The authors go on to show that the ACF complex and Ino80 are also dependent on the acidic patch for remodeling. Some points that need to be addressed.We are glad the reviewer finds the work exciting and thank the reviewer for the helpful comments and suggestions, which we address below.1) This paper needs to include some biochemical characterization of the APM nucleosomes compared to WT. Are they as stable for example at higher salt, etc. While the location of the mutations suggests they should not affect nucleosome stability, this can't be assumed. Moreover, the fact that Ino80 remodeling is also affected by the mutations without having similar Auto inhibitory domains suggest that the effect of the mutations is on the nucleosome substrate rather than through the remodelers.

The reviewer raises a good point. To test for stability effects we have now assessed salt dependent dissociation of the nucleosome. We find no large differences in the dissociation of WT vs acidic patch mutant (APM) nucleosomes as a function of increasing salt (Author response image 1). In addition, since we can rescue the majority of the APM defect with mutations in AutoN and now NegC, we interpret this to mean that the main effect of the acidic patch mutation is not due to gross defects in the nucleosome substrate. Furthermore, CHD1 can also efficiently remodel APM nucleosomes (Levendosky et al., 2016) suggesting that these mutations do not grossly destabilize nucleosomes. We also recognize the reviewer’s point about the effects on INO80. However, given that many nucleosome binding proteins have been shown to directly contact the acidic patch we speculate that a specific region of the INO80 complex contacts the acidic patch during remodeling. We anticipate that future work will identify this region of INO80.

**Author response image 1. respfig1:** Salt-based nucleosome dissociation of WT and APM nucleosomes. A. 15 nM 0/60 cy3-DNA labeled Nucleosomes were incubated in 25 mM HEPES pH 7.5 and varing concentrations of NaCl at 20ºC and resolved on a 6% native polyacrylamide gel. The gels were scanned for the cy3 label (green symbol on nucleosome and DNA cartoons on left of gel). B. Quantification of the fraction of nucleosomes remaining as a function of salt concentration. Fraction nucleosomes was determined by taking the ratio of the signal for the nucleosome bands to the signal for all the bands in each lane. Values were normalized to unincubated controls. The points and error bars plotted reflect the mean and standard error for n=3 experiments.

2) The assumption that the single point mutant is simply a weaker version of the 4-point mutants in the acidic patch may not be true. It may act completely differently. This assumption would be stronger if additional single point mutants were examined.

We thank the reviewer for this suggestion as the experiment suggested by the reviewer has yielded a new insight. If the acidic patch residues act independently we would expect the individual mutations to have smaller defects than the combination of mutations and expect the individual defects to be additive. In contrast if the acidic patch residues act cooperatively, we would expect the individual mutations to have comparable defects as the combination of mutations. Interestingly, as shown in Figure 1—figure supplement 3, except for E64A, all the single alanine mutants have comparable defects as the four-point mutants combined. This observation suggests that 3 out of the 4 acidic patch residues act cooperatively.

3) The mechanism would be strengthened if the authors can demonstrate a direct interaction between AutoN and the acidic patch and that this is reduced in the mutants. Simple peptide pulldown assays with Auto N compared to the H4 tails would suffice.

The reviewer makes a very valid suggestion. We did attempt to investigate by pull-down assays whether an AutoN peptide containing the 2R residues directly interacts with acidic patch nucleosomes but were unable to observe detectable binding. A crystal structure of the relevant AutoN region has been published showing that the 2R residues are within a loop region (Yan et al., 2016). We speculate that if the 2R residues are directly interacting with the acidic patch, this interaction may require the surrounding structured regions of AutoN for conformationally constraining the 2R residues in the loop.

However, to address the general point made by the reviewer of direct interaction we have carried out a cross-linking mass spec analysis in the presence of ADP and ADP-BeF_x_. We used the zero-length crosslinking system of EDC-NHS, which catalyzes amide bond formation between acidic residues and lysines. Substantial previous work has suggested that ADP-BeF_x_ mimics an activated state of the SNF2hnucleosome complex. We therefore determined the acidic patch cross-links that were specific to the ADP-BeF_x_ state. This approach identified interactions between the acidic patch and both AutoN and the NegC region of SNF2h as being highly enriched in the ADP•BeFx state (Figure 2B). This comparative cross-linking approach also identified another domain interaction that is enriched in the ADP•BeFx state and is consistent with an activated state, namely the H4-tail binding one of the SNF2h RecA domains.

Additionally, we now show that mutagenesis of NegC rescues the defect of the acidic patch to a greater extent than the 2R residues in AutoN (Figure 2). These results are consistent with the interaction between NegC and the acidic patch playing a mechanistically significant role. This data does not rule out a direct interaction with the 2R residues in AutoN because the cross-linking chemistry cannot generate cross-links between Arginines and acidic residues. However, the new data is consistent with a more complex model than we had previously proposed. We now propose a revised model, in which the acidic patch promotes a relief of auto-inhibition by both the NegC region and the AutoN region.

We are very grateful to the reviewer for suggesting that we look for a direct interaction as the data from this search has led to a more sophisticated mechanistic model.

4) The authors need to make the case that studying hSn2h in the absence of its partner proteins is warranted. The fact that ACF (of which hSf2h is a subunit) is much less dependent on the acidic patch than hSn2h alone could be construed as suggesting these are artifacts of studying hSnf2h out of its normal context.

We acknowledge the reviewer’s skepticism about looking at the SNF2h protein alone. However, we believe that studying how the core ATPase functions can help better understand how additional subunits like Acf1 regulate the intrinsic properties of SNF2h. A study by the Muir group that was published in Nature after we received the first round of *eLife* reviews also shows smaller effects of the acidic patch with ACF compared to SNF2h. However, they report that with other SNF2h complexes such as RSF and NURF the effect of mutating the acidic patch is comparable to that with SNF2h. Thus, we view the effects on SNF2h as a mechanistically meaningful baseline. We note that the Muir study does not provide a mechanistic explanation for the acidic patch effects. Our revised work directly addresses the mechanistic basis by showing how the acidic patch relieves autoinhibition in SNF2h. The model we propose for SNF2h alone can provide a foundation for future studies to understand why other SNF2hcontaining complexes are more sensitive to acidic patch mutations than the ACF complex.

Reviewer 2The main findings of this work are that the ISWI and INO80 remodelers are sensitive to the acidic patch on histone H2A. The ISWI ATPase subunit, SNF2h, is more extensively studied, where it is shown that nucleosome sliding is dramatically reduced with a quadruple acidic patch mutant. A nice complementary experiment is the competition with the LANA peptide, which is known to bind to the acidic patch, and therefore serves as a confirmation that this region is important for nucleosome sliding by ISWI. By using saturating amounts of remodeler, the authors demonstrate that the defects in sliding are not solely due to binding (though nucleosome interactions are decreased with the mutants), and therefore the defect is catalytic in nature. The authors find that a common mutant of the autoinhibitory "auto-N" motif (2RA) can rescue the remodeling defects, and they hypothesize a direct connection between basic region of the auto-N and acidic patch of H2A.1) While I agree with the general conclusions, the single molecule experiments presented in Figure 3 are harder to interpret. The authors propose that the E64R mutant increases the pause durations (which would be similar to what was observed for H4∆tail and short DNA linker nucleosomes in smFRET experiments; see Hwang et al., 2014). The author report that nucleosomes are shifted in steps, as expected for ISWI, and also the intriguing possibility that the E64R mutant alters the step size. A significant concern of mine, however, is that the example traces showing steps (e.g. Figure 3B) were rather noisy, and I would like to see more examples of the steps (as a supplement).

We are glad the reviewer agrees with our general conclusions and address the reviewer’s concerns about the single-molecule experiments below.

As the reviewer suggested, we have added additional example traces in a new supplemental figure, Figure 3—figure supplement 5. We believe one reason the E64R trace in the original version of Figure 3 in particular appears noisy is because of the very long-time scale on the x-axis compared to the other example traces in Figure 3B. In addition to providing more example traces, we have replaced some of the example traces in Figure 3B with ones that span, as much as possible, roughly equivalent durations, so that they have similar x-axes and are more visually comparable.

We note that all of the example traces in Figure 3B and in Figure 3—figure supplement 5 are smoothed for visualization only (not for analysis). The apparently noisiness of example traces are subject to effects of the smoothing filter used, the linewidth used to plot the data, the scaling of the axes, and so forth. We have plotted our example traces with a fairly conservative smoothing filter so as not to obscure features in the data. In addition, it is usually possible to decrease the noise in smFRET data by increasing the power of the imaging laser, with a trade-off in terms of faster photobleaching. However, in our case we kept the laser as low as possible, while still maintaining enough signal to noise that our HMM algorithm could detect pauses (see below), because the extremely slow remodeling with the E64R nucleosomes necessitated keeping the photobleaching rate as low as possible. Even so, as noted in the text, we cannot capture all of the remodeling reaction with E64R, because the FRET dyes often photobleach faster than SNF2h can remodel the E64R nucleosomes out of FRET range. For all of these reasons, our example traces may appear noisier than traces in other published smFRET work.

The reviewer’s concern about the noisiness of the traces may reflect a concern that our HMM fitting routine is unable to robustly detect steps, particularly in the E64R data, where the smaller distances translocated between pauses means the pause states are at closer FRET values to one another. This concern is a valid one. To address the effect of noise on the robustness of our HMM fitting algorithm (pyhsmm), we added synthetic noise to the smFRET data with WT SNF2h and the E64R nucleosomes. As shown in Author response image 2, 50% more noise in the E64R/WT data does not, within error, affect either the durations of the pauses, or their location (as reflected by how far the nucleosome is moved between pauses), as quantified by pyhsmm. When the noise is doubled, as in Author response image 2, the outcome of pyhsmm is still largely comparable to the original data, although some pauses are missed (as indicated by the longer p1 pause duration and larger step size between wait and p1). pyhsmm is thus extremely robust, even with noise levels significantly higher than those in our data.

**Author response image 2. respfig2:** The addition of synthetic noise does not significantly change the outcome of our HMM fitting routine. Synthetic, uncorrelated, Gaussian-distributed noise was added independently to the Cy3 and Cy5 intensities for each E64R/WT trace, and then the pyhsmm analysis was re-run on these noisier data. Pause durations and changes in nucleosome positions (step sizes) were then re-computed and are here plotted as in Figure 3 in the main text. In A and B, the standard deviation of the added Gaussian noise was 0.4; in C and D, the standard deviation of the added noise was 0.6. The standard deviations of the original Cy3 and Cy5 intensity data were on average 0.5 and 0.3 respectively, so the data in A and B have about 50% more noise than the original data, while the data in C and D have about double the noise of the original data.

Also, the HMM step sizes should agree with step sizes that one can obtain from the step finding algorithm published by Kerssemakers et al., (2006).

In Author response image 3 we show the results of the Kerssemakers algorithm for several example traces with WT SNF2h and WT nucleosomes, compared to the outcome of our pyhsmm hidden Markov model analysis. Overall the two algorithms are in good agreement. However, the Kerssemakers algorithm, which was originally designed for optical trapping data (that is, one-dimensional data), works only with 6 the FRET values for each trace. Our HMM jointly fits the Cy3 and Cy5 intensity data, which means it uses more information to find steps, which allows it to more accurately detect real steps and avoid false positives. Where the results of the Kerssemakers algorithm and pyhsmm differ, as in the bottom right trace in Author response image 3, an inspection of the Cy3 and Cy5 intensity traces show that pyhsmm fits the Cy3 and Cy5 intensities well.

We note that all of the fits obtained from our HMM algorithm were visually inspected and confirmed to be reasonable and have added a clarification to the Materials and methods section to this effect.

**Author response image 3. respfig3:** Comparison of the fit obtained by our HMM algorithm (pyhsmm) to that obtained by the Kerssemakers algorithm. Cy5 intensities (red data), Cy3 intensities (green data), and FRET values (blue data) are shown for 4 of the example traces in Figure 3 and Figure 3—figure supplement 5. Data are plotted as in those figures (including a 0.95-second smoothing filter, for visualization only). Dashed red and green lines in top panels show fits generated by pyhsmm to the Cy3 and Cy5 intensity data; dashed gray lines in the bottom panels show pyhsmm results in terms of FRET. The Kerssemakers algorithm acts only on the FRET values, so the outcome of Kerssemakers is shown on the bottom FRET panels only, as solid black lines.

If the authors assemble the steps into histograms based on FRET values at each step, it should be apparent what the preferential steps are and whether they change with AP nucleosomes. Such analysis would be more informative and appropriate than the cumulative probability plots shown in Figure 3E.

We acknowledge the reviewer’s suggestion for generating histograms based on FRET values, but we prefer to display the same information in terms of CDFs based on basepairs moved, for two reasons discussed in more detail below:

**Author response image 4. respfig4:** Histograms of FRET values for the p1 and p2 pauses for WT SNF2h with WT nucleosomes. A. Bin positions as in Blosser et al., 2009 (every 0.04 FRET, starting at 0 FRET). B. Bin positions every 0.05 FRET, starting at 0.04 FRET. Due to the non-linearity of our calibration curve, FRET values of pauses are not representative of preferential step locations. We show histograms of FRET values here simply to illustrate the effect of bin choice on the way the data appear, which we chose to do with reference to previously published histograms of FRET values from Blosser et al.,2009.

1) Due to the non-linearity of our calibration curve, we feel that describing preferential steps in terms of FRET values will be misleading.

2) We prefer cumulative density functions (CDFs) to histograms, because where peaks or clusters of data appear in histograms is highly sensitive to choice of bin width and position (see Author response image 4; for reference, see also the Discussion section at http://www.andata.at/en/software-blog-reader/why-we-love-the-cdfand-do-not-like-histograms-that-much.html). CDFs provide a more quantitative way to compare distributions obtained under different conditions (like WT vs E64R nucleosomes), by avoiding potential artifacts generated by the binning procedure in making histograms.

With regards to analyzing steps in terms of FRET values: in previous work observing ISWI enzymes remodeling single nucleosomes (e.g. Blosser et al., 2009, Deindl et al., 2013), the relationship between FRET and bp was well-described by a linear approximation, as determined by a calibration curve obtained the way ours was (see Figure 3—figure supplement 4). Thus in, for example, Blosser et al., 2009, an initial 7 bp step resulted in a change in FRET that was twice as large as a subsequent 3 bp step; that is, an initial step twice as large in bp resulted in a change in FRET twice as large as well. This linear relationship between FRET and bp explains why the first peak in Blosser et al. 2009 Figure 3b is at ~0.55 FRET (whereas in our system it is at ~0.75 FRET). In contrast, our calibration curve demonstrates that FRET and bp are not linearly related but rather follow the expected R^-6^ relationship, with our nucleosomes starting outside of the pseudo-linear regime. Nucleosomes in a single data set will not all start at exactly the same FRET value, and since they start outside the pseudo-linear range, even if they are moved the same number of basepairs by SNF2h, they will not all have the same change in FRET. So, to compare nucleosomes both within a dataset and between datasets we converted the steps to bp first. In our system, the initial ~7 bp step and the subsequent ~3 bp step both lead to FRET changes of similar magnitudes, because of how these steps move the nucleosome along the calibration curve. We believe our FRET calibration curve differs from that in Blosser et al., 2009 in part because we use a different labeling scheme where histone H3 is labeled instead of histone H2A and because the chemistry of the dye attachment to DNA differs from Blosser et al.

With regards to histograms versus CDFs: as an example of the sensitivity of histograms to bin width and position, in Figure R4 we compare histograms of FRET values for the p1 and p2 pauses for WT SNF2h with WT nucleosomes, using two different choices for the bins. In A, we plot our data as in Blosser et al., 2009, Figure 3B, in which the initial FRET values (i.e. the FRET values of the wait pause) are excluded, and p1 and p2 pause FRET values are binned at every 0.04 FRET. The data do not clearly fall into Gaussian-like peaks. However, in Author response image 4, we have chosen bin widths and locations that make two Gaussian-like peaks appear in the data. We do believe these particular peaks represent two real underlying clusters of FRET values in the data, based on the CDF analysis. However, it is very difficult to determine bin sizes and locations that reflect a best guess of the “true” clusters in the data from histogram analyses alone.

Because the step size of ISWI-family remodelers has been previously established in the literature (Blosser et al., 2009, Deindl et al., 2013), in our original manuscript we focused on comparing distributions of step sizes obtained with the different SNF2h and nucleosomal constructs. For such quantitative comparisons between data sets, we turned to the cumulative distribution function, or CDF, which does not suffer from the same binning pitfalls as histograms. A CDF represents the same information as a histogram, but without any smoothing or binning of the data. Peaks or clusters in the data appear as steep slopes in the CDF, whereas dips or separations between clusters appear as flat portions (see Figure 3—figure supplement 4 and the new Figure 3—figure supplement 6).

While the average step size (as well as other nuances of the step size distribution data) can be read off of the CDFs, the CDFs in Figure 3E emphasize differences between distributions of step sizes for different data sets, rather than the average step size for any particular data set. To emphasize the preferred step size for each condition, we have added a new supplemental figure, Figure 3—figure supplement 6, which represents the same step-size data in Figure 3E in three different visualizations: Kernel Density Estimation Plots (KDEs), CDFs and histograms. In that figure we also report the average step size for each data set, with the step size for WT SNF2h and WT nucleosomes in good agreement with step sizes reported for the ACF complex and other ISWI family remodelers.

2) On a related point, in the Discussion section, the description of sliding the E64R mutant is confusing with regard to increased pausing frequencies. The authors describe an apparent change in the stepsize in nucleosome sliding, which may or may not result from increased frequency of pausing, but no data has been presented for pausing frequency (distinct from pausing duration). Figure 3E is discussed with regard to the distances translocated, not on frequencies of pausing, so this point is unclear.

We thank the reviewer for pointing out this inconsistency in our terminology and have clarified the text by removing reference to pausing frequency. We now describe remodeling with the E64R nucleosomes resulting in shorter distances translocated between pauses, resulting in more pauses encountered per distance translocated.

3) Based on the core thesis of their work, that the acidic patch relieves inhibition by auto-N and therefore accelerates overall sliding, I think it would be worthwhile to investigate sliding behavior of H2AZ nucleosomes, which are known to have a more extensive acidic patch region. The Kingston lab (Goldman et al., 2010) showed that ISWI is more active with H2AZ nucleosomes. They found that the "additional acidic residues" of H2AZ were required for the higher sliding rate, but somewhat unexpectedly, adding just the additional acidic residues of H2AZ to normal H2A did not appear to alter sliding. While such experiments may not lend new insight for ISWI, H2AZ is a natural substrate for INO80 and therefore it would be worthwhile to compare sliding rates for H2A, H2AZ, and H2AZ-NK (acidic patch=normal H2A).

The reviewer raises a good point. The work from the Kingston lab is consistent with our current model for how the acidic patch relieves auto-inhibition by SNF2h. The comparable INO80 experiment suggested by the reviewer has recently been carried out by the Bartholomew lab (Brahma et al., 2017). They find that INO80 preferentially slides H2AZ nucleosomes over H2A nucleosomes.

4) Figure 3—figure supplement 3A: The authors state that 2-dye nucleosomes were excluded from analysis, so that the FRET differences from distal and proximal dyes could be properly accounted for. This is fine, but it is unclear how both dyes together affect the resulting initial FRET. Was there some mechanism for ensuring that traces used for the remodeling experiments were also from single-dye nucleosomes?

We thank the reviewer for pointing to an aspect of our analysis that we did not properly clarify. Specifically, the same process by which two-dye nucleosomes were excluded from the FRET kernel density plots, namely, the exclusion of trajectories with two-step photobleaching events, was also used to exclude such nucleosomes from remodeling experiments. All movies of remodeling were of such duration as to ensure that all dyes photobleached before the end of the movie, so that we could be sure to exclude any nucleosomes with two Cy3 dyes. Moreover, in contrast to earlier work (e.g. Blosser et al., 2009), we used a 2:1 unlabeled H3:labeled H3, instead of a 1:1 ratio, resulting in very few doubly-labeled nucleosomes. We have clarified the text accordingly.

5) Figure 3—figure supplement 5: why are the heights of the peaks from the two differently positioned dyes so different. Shouldn't these have similar densities? The 12/78 seems somewhat odd with the closer dye yielding a much lower peak than the more distal dye. Can the authors comment on this?

The reviewer is correct that, in theory, nucleosome assembly should result in roughly equal proportions of proximally labeled and distally labeled nucleosomes. However, in our hands nucleosomes assemble with a preference for the H3 histone label to be away from the DNA label. Unequal assembly preference has also been observed by others (e.g., Qiu et al., 2017).

With regards to the 12/78 calibration data in particular: we removed this point and then repeated the calibration curve fit, and obtained similar fit parameters as when the 12/78 data were included (without 12/78, R_0_ = 11.3 ± 1.1 nm, d_0,prox_ = 5.9 ± 1.1 nm, θ_prox_ = 180.0 ± 19.5°, d_0,dist_ = 11.0 ± 1.0 nm, θ_dist_ = 89.1 ± 6.9°, compared to R_0_ = 10.9 ± 1.1 nm, d_0,prox_ = 5.8 ± 1.0 nm, θ_prox_ = 153.8 ± 20.6°, d_0,dist_ = 10.6 ± 1.0 nm, θ_dist_ = 87.3 ± 12.0° in the text).

6) The authors state in the Discussion section "…acidic patch mutations increase both the durations and frequencies of pauses." The authors only present the E64R mutant in smFRET experiments, and so should either present data for other mutants or adjust this sentence accordingly.

We thank the reviewer for pointing out this lack of clarity. We now state that we use the smFRET data with E64R to infer the role of the acidic patch.

Reviewer 3In this manuscript it is shown that mutation of residues that comprise the considered acidic patch on the nucleosome affect the ability of some, chromatin remodeling enzymes to act on nucleosomes bearing these nucleosomes. This is interesting as this region of the nucleosome has previously been shown to act as a binding interface for other types of nucleosome binding proteins. However, the acidic patch is not universally required for remodeling enzymes as the enzyme Chd1 has previously been shown to act on nucleosomes where this interface is altered. Also, when SNF2h is present as part of the ACF complex the effect of the AP region is reduced 20-fold.The functional significance of this interaction is investigated further by studying the rate at which a more active mutant of the SNF2h enzyme (AutoN) remodels nucleosomes. This mutant enzyme is found to be less sensitive to alterations to the acidic patch on the nucleosome. As a result, a model is proposed in which AutoN interacts with the acidic patch in the active state, but switches back to an inhibitory conformation during cycles of enzyme activity. Support for this is gained from single molecule studies showing that pausing is increased by mutation of the AP surface. Although the AutoN mutation does not affect pausing on normal nucleosomes, it does act to restore normal enzyme action on AP mutant nucleosomes.It's known that the AutoN loop interacts with ATPase lobe 2 and that the histone H4 competes for engagement with this region (Yan et al. 2016). Here it is proposed that the acidic patch region acts as an alternative location via which both AutoN and the H4 tail can interact. A major weakness of the manuscript is that no evidence is presented to directly support direct association of these regions. Ideally, dynamic association of AutoN and the H4 tail with the AP region would be directly measured during the reaction cycle. The authors have established a single molecule FRET system that could be used to make this type of measurement, it would none the less be a substantial amount of new work to include this type of work.

We thank the reviewer for their comments and acknowledge that evidence for a direct contact with the acidic patch and AutoN would strengthen the manuscript. We describe below how we have addressed these and other concerns.

Some form of evidence should be obtained to indicate that AutoN and the AP region interact directly. There are a number of different approaches that could be used for this including FRET, X-link MS or pull down. This would provide important evidence supporting the model presented in Figure 5.

The reviewer makes a very valid suggestion. We did attempt to investigate by pull-down assays whether an AutoN peptide containing the 2R residues directly interacts with acidic patch nucleosomes but were unable to observe detectable binding. A crystal structure of the relevant AutoN region has been published showing that the 2R residues are within a loop region (Yan et al., 2016). We speculate that if the 2R residues are directly interacting with the acidic patch, this interaction may require the surrounding structured regions of AutoN for conformationally constraining the 2R residues in the loop.

However, to address the general point made by the reviewer of direct interaction we have carried out a cross-linking mass spec analysis in the presence of ADP and ADP-BeF_x_. We used the zero-length crosslinking system of EDC-NHS, which catalyzes amide bond formation between acidic residues and lysines. Substantial previous work has suggested that ADP-BeF_x_ mimics an activated state of the SNF2-hnucleosome complex. We therefore determined the acidic patch cross-links that were specific to the ADP-BeF_x_ state. This approach identified interactions between the acidic patch and both AutoN and the NegC region of SNF2h as being highly enriched in the ADP•BeFx state (Figure 2B). This comparative cross-linking approach also identified another domain interaction that is enriched in the ADP•BeFx state and is consistent with an activated state, namely the H4-tail binding one of the SNF2h RecA domains.

Additionally, we now show that mutagenesis of NegC rescues the defect of the acidic patch to a greater extent than the 2R residues in AutoN (Figure 2). These results are consistent with the interaction between NegC and the acidic patch playing a mechanistically significant role. This data does not rule out a direct interaction with the 2R residues in AutoN because the cross-linking chemistry cannot generate cross-links between Arginines and acidic residues. However, the new data is consistent with a more complex model than we had previously proposed. We now propose a revised model, in which the acidic patch promotes a relief of auto-inhibition by both the NegC region and the AutoN region.

We are very grateful to the reviewer for suggesting that we look for a direct interaction as the data from this search has led to a more sophisticated mechanistic model.

The single molecule FRET measurements appear to indicate different lengths of DNA transit over the nucleosome, and that there may be differences from previously published data. This is potentially very interesting.

We believe the reviewer is referring to the difference in the distances the nucleosome is moved with each translocation event for E64R versus WT nucleosomes, which we agree is quite interesting. However, we would like to clarify that we find the distance the nucleosome is moved with WT SNF2h and WT nucleosomes to be consistent with previously published data. We have edited the text to make this point more clear and have included a new Figure 3—figure supplement 7 that addresses this point (see also below).

However, the noise in the individual traces appears to be high.

We believe one reason the E64R trace in the original version of Figure 3 in particular appears noisy is because of the very long-time scale on the x-axis compared to the other example traces in Figure 3B. In addition to providing more example traces, we have replaced some of the example traces in Figure 3B with ones that span, as much as possible, roughly equivalent durations, so that they have similar x-axes and are more visually comparable.

We note that all of the example traces in Figure 3B and in Figure 3—figure supplement 6 are smoothed for visualization only (not for analysis). The apparently noisiness of example traces are subject to effects of the smoothing filter used, the linewidth used to plot the data, the scaling of the axes, and so forth. We have plotted our example traces with a fairly conservative smoothing filter so as not to obscure features in the data. In addition, it is usually possible to decrease the noise in smFRET data by increasing the power of the imaging laser, with a trade-off in terms of faster photobleaching. However, in our case we kept the laser as low as possible, while still maintaining enough signal to noise that our HMM algorithm could detect pauses (see below), because the extremely slow remodeling with the E64R nucleosomes necessitated keeping the photobleaching rate as low as possible. Even so, as noted in the text, we cannot capture all of the remodeling reaction with E64R, because the FRET dyes often photobleach faster than SNF2h can remodel the E64R nucleosomes out of FRET range. For all of these reasons, our example traces may appear noisier than traces in other published smFRET work.

The reviewer’s concern about the noisiness of the traces may also reflect a concern that our HMM fitting routine is unable to robustly detect steps, particularly in the E64R data, where the smaller distances translocated between pauses means the pause states are at closer FRET values to one another. This concern is a valid one. To address the effect of noise on the robustness of our HMM fitting algorithm (pyhsmm), we added synthetic noise to the smFRET data with WT SNF2h and the E64R nucleosomes. As shown in Author response image 2, 50% more noise in the E64R/WT data does not, within error, affect either the durations of the pauses, or their location (as reflected by how far the nucleosome is moved between pauses), as quantified by pyhsmm. When the noise is doubled, as in Author response image 2, the outcome of pyhsmm is still largely comparable to the original data, although some pauses are missed (as indicated by the longer p1 pause duration and larger step size between wait and p1). pyhsmm is thus extremely robust, even with noise levels significantly higher than those in our data.

The differences in the movements and the errors associated with the measurements should be established rigorously.

We now more clearly explain in the methods how these errors were established. Specifically, the errors associated with our measurement of how far the nucleosome is moved with each translocation event were established by a bootstrapping procedure and are shown as shaded regions in the CDF plots of nucleosome movements (Figure 3E). These shaded reasons provide an estimate of the uncertainty in our measurement of the entire distribution of step sizes for each condition and show that the entire distribution of step sizes for the E64R nucleosomes with WT SNF2h are shifted to smaller step sizes, by a statistically significant amount.

We have also added a new Figure 3—figure supplement 6 which reports the mean step size and a standard error on that mean for each set of conditions. These mean step sizes similarly show that on average, the nucleosome is translocated a shorter distance with E64R nucleosomes and WT SNF2h (5.6 ± 0.3 bp for the first step) compared to WT SNF2h with WT nucleosomes (7.8 ± 0.3 bp). However, as shown in Figure 3-S7, the mean step size values alone do not do justice to how different the distributions of step sizes are with E64R versus WT nucleosomes, which is better reflected in the KDEs in Figure 3—figure supplement 7 or the CDFs in Figure 3E.

As shown in Figure 3—figure supplement 6, the step sizes we measure for WT SNF2h and WT nucleosomes are comparable to those reported in the literature for other ISWI family remodelers (Blosser et al., 2009, Deindl et al., 2013, Hwang et al., 2014). The small differences in measured step size compared to the literature likely reflect differences between our approach to determining step size compared to previous approaches in the literature. Specifically, in Blosser et al., 2009, Deindl et al., 2013 and Hwang et al., 2014, the FRET values for all pauses in all traces were combined into a single histogram. Several Gaussians were then fit to this histogram, and the differences between means of these Gaussians, converted from FRET to bp, were taken to be the sizes of each step. As noted above in a response to reviewer 2, where peaks appear in histograms—and therefore where Gaussians will be fitted—is strongly dependent on the bin sizes and locations used. Moreover, combining all pauses into one histogram means data for each pause may overlap. Here, we have instead separated out each pause (wait, p1, p2), and computed the change in bp between the first and second pauses, and between the second and third pauses. This allows us to report the mean step size as the mean value of each data set, rather than fitting a Gaussian to the decidedly non-Gaussian distributions in Figure 3—figure supplement 6.

The authors should also discuss whether the differences in DNA movements are likely to involve differences in the rate at which elementary steps are generated, or the time during which SNF2H is committed to single base steps.

The reviewer raises a good point which will help us clarify our discussion of the translocation events. Assuming SNF2h, like yeast ISWI remodelers (see Deindl et al., 2013), translocates the nucleosome in elementary steps of 1 or 2 bp each, the differences in DNA movements quantified in Figure 3E reflect how many of these elementary steps SNF2h takes during a translocation phase, before entering a new pause phase. Our data suggest that with E64R nucleosomes, SNF2h takes fewer elementary steps before entering a new pause phase. We have clarified this point in the Results section that discusses the step size. Specifically, we say: “Given that ISWI family remodelers have been shown to translocate a nucleosome in elementary steps of 1-2 bp (Deindl et al., 2013), our results suggest that with E64R nucleosomes, SNF2h takes fewer of these elementary steps in succession during the translocation phase, before entering a new pause phase.”

[Editors’ note: the author responses to the re-review follow.]

1) The EDC cross-linking/mass spec experiments provided new and interesting information regarding how Snf2h binds to the nucleosome. Some of those data, however, appeared to be somewhat selectively interpreted, biased toward the authors' model. While cross-linking of the AutoN domain was observed with the acidic patch, cross-links were also observed for other regions of Snf2h, such as the HAND/SANT/SLIDE (HSS) domain and RecA domains. In addition, a significant number of intramolecular cross-links were observed from the AutoN to other parts of Snf2h. Cross-links closest to 2RA are to other parts of Snf2h rather than the acidic patch. The authors should discuss the potential significance of these interactions. It would be helpful if the authors could devise a figure to more precisely indicate locations of cross-links with respect to 2RA and NegC.

The reviewers raise some excellent points. Before we address these points we’d first like to clarify that the heat map in Figure 2B is a mainly a visual aid to summarize two complex data sets and to highlight the major changes in domain interactions between ADP and ADP-BeF_x_ nucleosome-SNF2h complexes in a way that is qualitatively intuitive. Abstracting and displaying the data this way however introduces some unavoidable distortions. For example, there are 8 crosslinked spectral counts identifying AutoN to H2A-Acidic Patch interaction in ADP-BeF_x_ and 0 with ADP. Because zero values would lead to infinite ratios, we arbitrarily assign 1 count to the ADP-BeF_x_ interaction. Normalization (which was conservative, see response to point 2), and the choice of 4-fold enrichment as the breakpoint for shading the darkest red color results in this interaction appearing less important than other domain-domain interactions which are colored more strongly. However, the underlying ratio is technically infinite. Hence, we also include the figure supplement, which shows a representation that is closer to the raw data with each dot representing a pair of crosslinked residues between the acidic patch and SNF2h.

In terms of the comment pertaining to the cross-links closest to 2RA, the residues nearest to these two arginines (R142 and R144) do actually crosslink directly to the H2A acidic patch. However, we realized that the original figure supplement did not easily allow an assessment of this issue. We have now added red lines to the figure supplement (Figure 2—figure supplement 4) to explicitly show the positions of R142 and R144 within the AutoN region of SNF2h and mentioned this clarification in the main text (subsection “The AutoN and NegC regions of SNF2h cooperate with the acidic patch to enable maximal remodeling”). Comparing the ADP and ADP-BeF_x_ grids now more clearly shows that the residues nearest 2RA also crosslink directly to the H2A acidic patch, and only do so in the present of ADP-BeF_x_ (left two panels of the supplement). We did not highlight NegC in the figure supplement, because the NegC domain is already demarcated by the domain boundaries on the right side of this figure (Figure 2—figure supplement 4).

In terms of the comment about intramolecular cross-links from the AutoN to other parts of Snf2h, it appears that our data representation may have miscommunicated the main point we were trying to make. While the reviewers are correct in that there are intra-molecular interactions suggested for the ADP and ADP-BeF_x_ states separately, when we ask which AutoN interactions are significantly enriched with ADP-BeF_x_, these are to the acidic patch and to other parts of the nucleosome as shown in Figure 2B. AutoN to SNF2h crosslinks are otherwise light colored in the heatmap. We have now added a clarification sentence about these comparisons in the main text (subsection “The AutoN and NegC regions of SNF2h cooperate with the acidic patch to enable maximal remodeling”, Figure 2).

In terms of other SNF2h regions cross-linking to the acidic patch in the ADP-BeF_x_ state, the reviewers are correct that such interactions are apparent in the data, for instance between SNF2h-HSS and the H2B acidic patch and RecA domains to the H2A acidic patch as well as other histones. As suggested by the reviewers we have expanded our discussion of the crosslinking data to include explanations for the potential significance of these cross-links (subsection “The AutoN and NegC regions of SNF2h cooperate with the acidic patch to enable maximal remodeling”).

2) There were many fewer cross-links in the ADP compared to the ADP-BeF_x_ complex, which makes it unclear whether the ratios shown in Figure 2B are the most appropriate way to present nucleotide-dependent differences. Were those data normalized? If not, could a set of cross-links not expected to change in response to nucleotide (e.g. Snf2h intradomain, or histone-histone) be used as an internal standard for normalizing the ADP and ADP-BeF_x_ data?

The reviewers’ suggestion to normalize based on domain interaction that is not expected to change is a good one in principle. However, in practice it is difficult to identify any such interactions for several reasons. For instance, intra-domain interactions can’t be assumed not to change because there are two protomers of SNF2h present and intra-protein SNF2h-SNF2h crosslinks cannot be resolved from inter-protein SNF2h-SNF2h crosslinks with the mass spectrometry method used here (Figure 2 legend). Furthermore, we note that previous work from our group has shown that distortions to the histone core occur in the ADP-BeF_x_ state with SNF2h (Sinha et al., 2017) suggesting that intra-domain histone crosslinks cannot be assumed to remain constant across ATP states. That said, we note that the vast majority of the intra-domain SNF2h changes are not marked as significant using our method clarified further below (light colored tiles in the Figure 2B heat map).

Because we could not reliably normalize to an interaction not assumed to change, we chose a different approach of displaying the cross-linking data as shown in Figure 2B. This figure, which shows the crosslinking data aggregated into domain level interactions, is normalized such that the median Log2 value of spectral counts observed in ADP-BeF_x_ to ADP is centered on 0; e.g. the assumption being made is that most domain-domain interactions will remain the same between the two conditions, and that the differences in numbers of crosslink counts is primarily due to experimental handling and instrument performance. Note that the un-normalized median of the Log2 spectral count ratio is 1.03, and all that has been done is to center the distribution (histogram in Figure 2B) on a value of 0. Thus, the correction being made is conservative in terms of which domain interactions we identify as significantly enriched in the ADP-BeF_x_ condition.

In terms of the reviewers’ observation of fewer cross-links in the ADP state compared to the ADP-BeF_x_ state, it is possible that this is a result of the slightly lower (~2-fold, Leonard and Narlikar, 2015) binding affinity observed between SNF2h and mononucleosomes in this condition. However, such a difference does not change our interpretation of this figure (Figure 2) significantly. The function of this figure is to serve as a heuristic with which to identify domain interactions that are specific to the two mechanistic states of the complex. The identity of the most and least enriched domain interactions will not change based on the normalization. The tiles of the heat map that we chose to color most intensely (log2 ratio > 2) are sufficiently separated from the bulk of the distribution that we are confident that they reflect real changes in the domain interactions of the assembly

3)The authors should consider additional experiments to determine whether the HSS domain might be the element that directly interacts with the acidic patch. One experiment would be to test whether the 2RA mutant of the AutoN can rescue the Lys-to-Ala mutants in the HSS that the authors' found disrupt activity. A rescue would signify that AutoN bypass is downstream of HSS regulation. Alternatively, the HSS mutants that fail to slide nucleosomes could be compared to wt in EDC cross-linking/mass spec experiments. Selective loss of other HSS cross-links with the acidic patch would support a direct interaction. These experiments are not a requirement for publication, but if feasible on a short time scale could provide additional valuable information.

We agree with the reviewers that it is important to determine whether the HSS may be directly interacting with the H2A acidic patch. While we have not performed the specific experiments suggested by the reviewers, we do present new data from a different experiment that tests which crosslinks are dependent on direct interactions with the H2A acidic patch (Figure 2—figure supplement 5). We find that addition of the LANA peptide to the SNF2h-nucleosome complex in the ADP-BeF_x_ state significantly blocks acidic patch crosslinks to all regions of SNF2h except for the HSS. While this experiment does not rule out a direct HSS-acidic patch interaction, it suggests that the positioning of the HSS in the ADP-BeF_x_ state is not solely determined by direct binding to the H2A acidic patch residues contacted by the LANA peptide.